# Mechanisms and intraseasonal variability of the South Vietnam Upwelling, South China Sea: role of circulation, tides and rivers

Marine Herrmann[1], Thai To Duy[2], and Patrick Marsaleix[1]

[1]Université de Toulouse, LEGOS, IRD/CNRS/CNES/Université de Toulouse, Toulouse, France
[2]Institute of Oceanography (IO), Vietnam Academy of Science and Technology (VAST), Nha Trang, Vietnam

**Correspondence:** Marine Herrmann (marine.herrmann@ird.fr)

**Abstract.** Summer monsoon southwest wind induces the South Vietnam Upwelling (SVU) over four main areas along the southern and central Vietnamese coast: offshore the Mekong shelf (MKU), along the Southern and Northern coasts (SCU and NCU) and offshore (OFU). Previous studies have highlighted the roles of wind and Ocean Intrinsic Variability (OIV) in the SVU intraseasonal to interannual variability. The present study complements these results by examining the influence of tides and river discharges and investigates the physical mechanisms involved in MKU functioning.

MKU is driven by non chaotic processes, explaining its negligible intrinsic variability. It is triggered first by the interactions of currents over a marked topography. The surface convergence of currents over the southwestern slope of the Mekong shelf induces a downwelling of the warm northeastward alongshore current. It flows over the shelf and encounters a cold northwestward bottom current when reaching the northeastern slope. The associated bottom convergence and surface divergence lead to an upwelling of cold water which is entrained further north by the surface alongshore current.

Tides strengthen this circulation-topography-induced MKU through two processes. First, tidal currents weaken the current over the shallow coastal shelf by enhancing the bottom friction. This increases the horizontal velocity gradient hence the resulting surface convergence and divergence and the associated downwelling and upwelling. Second, they reinforce the surface cooling upstream and downstream the shelf through lateral and vertical tidal mixing. This tidal reinforcement explains 72% of MKU intensity on average over the summer, and is partly transmitted to SCU through advection. Tides do not significantly influence OFU and NCU intensity. Mekong waters slightly weaken MKU (by 9% on the annual average) by strengthening the stratification, but do not significantly influence OFU, NCU and SCU. Last, tides and rivers do not modify the chronology of upwelling for the four areas.

## 1 Introduction

The South Vietnam Upwelling (SVU) develops in the South China Sea (SCS) along and off the Vietnamese coast under the influence of summer monsoon southwest wind (Xie, 2003; Dippner et al., 2007). It is a major component of the SCS ocean circulation and influences the local climate (Xie, 2003; Zheng et al., 2016; Yu et al., 2020) as well as the biological activity and fishing resources (Bombar et al., 2010; Liu et al., 2012; Loisel et al., 2017; Loick-Wilde et al., 2017; Lu et al., 2018).

The SCS circulation along and off the central and southern Vietnamese coast is characterized by a southern anticyclonic circulation and a northern cyclonic circulation. They form an eddy dipole and are associated respectively with northeastward and southward boundary alongshore currents (Wyrtki, 1961; Wang et al., 2006a). Recent studies showed that SVU develops over four main areas (highlighted in Fig.1), driven by different mechanisms (Da et al., 2019; Ngo and Hsin, 2021; To Duy et al., 2022; Herrmann et al., 2023). The northeastward and southward boundary currents converge near $\sim 14°N$, forming the summertime eastward jet (SEJ). This offshore current separation induces the southern coastal component of the SVU, hereafter called SCU. Similar upwelling can develop along the northern coast (NCU), induced by the formation of seaward currents by the convergence of alongshore currents associated to small coastal eddies. Beside those Ekman transport-driven coastal upwellings, an Ekman pumping-driven offshore upwelling (OFU) develops north of the SEJ. Last, To Duy et al. (2022) and Herrmann et al. (2023) revealed the existence of an upwelling that develops over the Mekong shelf (MKU), whose physical mechanisms have yet to be explained.

The SVU shows a strong variability, from the daily and intraseasonal to the interannual scales. Wind is a major factor of this variability at different scales. Over the last two decades, numerous studies have showed the effects of wind and El Niño Southern Oscillation on the interannual variability of the large scale circulation formed by the SEJ and eddy dipole and of the SVU intensity (Kuo et al., 2004; Wang et al., 2006b; Li et al., 2014; Da et al., 2019; Ngo and Hsin, 2021; To Duy et al., 2022): stronger (weaker) summer monsoon southwest wind, enhanced (weakened) during La Niña (El Niño) events, induces more (less) intense SEJ and SVU than average. Only a few studies focused on the SVU variability at the intraseasonal scale. Isoguchi and Kawamura (2006) and Xie et al. (2007) showed that wind, itself strongly driven by Madden Jullian Oscillation at the intraseasonal summer scale, but also by tropical storms, is a major driver of SVU daily variability. Herrmann et al. (2023) more specifically showed that wind is the main driver of daily variability of SEJ and of SCU, OFU and MKU. They showed that NCU variability was rather driven by the strength of the large scale circulation at the intraseasonal scale: the well established circulation and southward boundary current during the core of summer prevent the development along the northern coast of seaward currents, hence of NCU. NCU rather develops at the beginning and end of the summer when this circulation is weaker, allowing small scale eddying structures and seaward currents to develop.

Recent modeling studies moreover revealed the high impact of ocean intrinsic variability (OIV) on the SVU. OIV is main related to the intense activity of (sub)mesoscale structures and eddies of strong chaotic nature in the SCS (Ni et al., 2021; Xiu et al., 2010). Li et al. (2014) first suggested that OIV could contribute to 20% of the SEJ interannual variability. Da et al. (2019) and To Duy et al. (2022) suggested that OIV influences the SVU interannual variability, in particular the OFU. To explain and quantify this impact of OIV, Herrmann et al. (2023) then performed and analysed an ensemble of high-resolution simulations for summer 2018, a year of strong SVU (Ngo and Hsin, 2021; To Duy et al., 2022). SCU and MKU are mainly driven by the strength of the SEJ, itself driven by wind intensity, and the impact of OIV is of second order: the ratio between the ensemble dispersion and average of upwelling intensity is smaller than 10%. This impact is slightly stronger for OFU (ratio of 18%) and significantly stronger for NCU (ratio of 37%). It is related to the spatial organisation of submesoscale to mesoscale structures

and eddies of strongly chaotic behavior, and to their interaction with the wind curl.

The Mekong shelf receives the large freshwater input from the Mekong river, with a monthly discharge varying between $15 \times 10^3$ and $25 \times 10^3$ m$^3$.s$^{-1}$ during the summertime (Chen et al., 2012). It also hosts a strong tidal activity. SCS is one of the few areas over the world where diurnal tides (mainly K1 and O1, with amplitudes $\sim$30-50 cm over the open sea region,

Guohong, 1986; Fang et al., 1999; Phan et al., 2019; Trinh et al., 2023) generally dominate semi-diurnal tides (mainly M2 and S2, with amplitudes generally smaller than 10-20 cm). M2 amplitude and tidal currents are however locally similar to K1 and O1 in the southwestern SCS, in particular over the Mekong delta shelf. There, the tidal amplitude of all components is stronger than over the rest of the basin (reaching $\sim$1 m for M2, see the estimations by FES2014b tidal atlas (Carrere et al., 2013) and numerical simulations in Trinh et al., 2023). The SVU may therefore be influenced by river water and tides, in particular in the

MKU and SCU regions, which host the Mekong plume and a strong tidal activity.

The role of the interactions between circulation, tides, rivers and topography was examined and demonstrated in other upwelling regions, mostly based on numerical experiments. In the northern SCS in particular, several authors examined upwellings developing off the coasts of Hainan Island. Lü et al. (2008) and Li et al. (2020) proved that tidal mixing induces

changes in horizontal pressure gradient, hence in circulation, which explains the development of upwelling off western and northeastern Hainan. The surface cooling is also intensified locally by the vertical tidal mixing of cold deep water with warmer shallow water (Li et al., 2020). Bai et al. (2020) moreover showed that the high-frequency variations of western Hainan upwelling were related both to horizontal advection due to tidal currents and to vertical velocity due to divergence/convergence triggered by tidal waves. Changes in horizontal density gradients and circulation induced by tidal mixing were also shown to

play a major role further north, in the upwelling that develops at the lee of a coastal promontory off the Yangtze River estuary in the East China Sea (Lü et al., 2006) and in the Bohai Sea (Xu et al., 2023). Lü et al. (2006) also mentioned the role of the Yangtze river discharge. Upwelling in those regions can moreover be induced by the interactions of geostrophic circulation (Lü et al., 2006).

Despite the importance of river discharge and tidal activity in the SVU area, very few numerical studies of SVU have been based on three-dimensional models including tides and realistic river discharges, and even fewer have investigated the impact of these processes. Among them, the process-oriented experiments of Chen et al. (2012) suggested that during summertime stratified conditions, tidal rectification, river discharge and local bathymetry could intensify the northern southward and southern northeastward currents and the associated SCU. In the 1/12° simulations of (Da et al., 2019) using monthly climatological

discharges for three rivers (Mekong, Pearl and Red rivers) but not including tides, rivers did not have a statistically significant effect on the global SVU intensity at the interannual scale. They did, however, produce strong differences for certain years. Their statistical significance should thus be explored given the OIV effect on the SVU.

This paper follows and belongs to the ensemble of studies cited above that investigated the functioning and variability of the upwelling that develops off the Vietnamese coast, in particular through its signature of sea surface temperature (SST). The general goal of the paper is to deepen our understanding of the functioning and variability of the SVU, considering potentially important processes and aspects which were not the focus of most of previous studies in this upwelling region: small scale dynamics and associated chaotic variability, tides and daily river discharges, upwelling over the Mekong shelf. This follows in particular the study of Herrmann et al. (2023), who quantified the impact of OIV at the intraseasonal scale based on ensemble simulations. Given the strong influence of OIV on SVU, we adopt the same probabilistic approach, following Waldman et al. (2017, 2018). We now aim first to quantify and explain the influence of tides and rivers on the SVU over its four areas of development at the daily to intraseasonal scales. Second, we aim to identify the physical mechanisms responsible for MKU development. For that, we perform high-resolution reference and sensitivity ensemble simulations using the same configuration of the SYMPHONIE model as To Duy et al. (2022) and Herrmann et al. (2023), including in particular realistic daily river discharges and explicit tide representation.

The model, reference and sensitivity ensemble simulations and statistical indicators are presented in Section 2. Results of the ensembles are statistically analysed in Section 3 to quantitatively assess the effects of tides and rivers on SCU, NCU, OFU and MKU intensity. Mechanisms behind MKU are explored in Section 4. The robustness of our conclusions is discussed in Section 5. Concluding remarks and future works are given in Section 6.

## 2 Methodology and tools

### 2.1 The ocean model SYMPHONIE

The SYMPHONIE model (Marsaleix et al., 2008, 2019) is a tridimensional ocean model developed by the SIROCCO group for the study of coastal and regional ocean, based on the Navier-Stockes primitive equations solved on an Arakawa curvilinear C-grid under the hydrostatic and Boussinesq approximations. Vertical mixing is parameterized according to the k-epsilon turbulence closure scheme (Rodi, 1987). The configuration used here was implemented by To Duy et al. (2022) over the Vietnamese coastal area and South China Sea region, using a standard horizontal polar grid with a resolution decreasing linearly from 1 km at the coast to 4.5 km offshore and 50 VQS (vanishing quasi-sigma) vertical levels. Fig. 1 shows the limits and bathymetry of the numerical domain. Forcings are the same as used by Herrmann et al. (2023) to perform an ensemble simulation over the period 2017-2018. The atmospheric forcing is computed from the bulk formulae of Large and Yeager (2004) using the 3-hourly output of the European Center for Medium-Range Weather Forecasts (ECMWF) 1/8° atmospheric analysis, distributed on http://www.ecmwf.int. Lateral ocean boundary conditions are provided by the daily outputs of the global ocean 1/12° analysis PSY4QV3R1 distributed by the Copernicus Marine and Environment Monitoring Service (CMEMS) on http://marine.copernicus.eu. The implementation of tides follows Pairaud et al. (2008, 2010), and we prescribe the same daily discharge of 36 rivers along the coast as Herrmann et al. (2023), including majors rivers as the Mekong and Red rivers.

The ability of the model to represent the ocean dynamics and water masses of the study area, from daily to interannual and from coastal to regional scales, was demonstrated in details by To Duy et al. (2022). For this, they compared a simulation performed over the period 2009-2018, hereafter called LONG, with satellite datasets of sea surface temperature, salinity and elevation and with in-situ temperature and salinity datasets from ARGO floats, glider, CTD and thermosalinograph measurements. In particular their high-resolution simulation, together and in agreement with a careful examination of available satellite and in-situ data, showed for the first time the existence of MKU, and the ability of the model to represent it.

## 2.2 Ensemble simulations

We performed three ensemble simulations over the period 2017-2018. The reference ensemble FULL includes rivers and tides and was already used by Herrmann et al. (2023) to examine the impact of OIV on the SVU intraseasonal variability. The additional sensitivity ensembles, NoRiver and NoTide, are identical to FULL, but respectively without river discharges and without tides. Each ensemble is composed of 10 members with perturbed initial small scale fields of temperature, salinity and currents, as detailed by Herrmann et al. (2023): the randomization strategy is based on the fact that most of the OIV develops at (sub)mesoscale related to eddies and structures of strongly chaotic behavior (Sérazin et al., 2015; Waldman et al., 2018) and smaller than $\sim 100$ km in the South China Sea (Lin et al., 2020; Ni et al., 2021). For each member M$XX$, with $XX$ going between 09 and 18, the initial state in terms of temperature, salinity, currents and sea surface elevation is the sum of a large scale state and a small scale state. The large scale state is the same for all members, given by the large scale state of January 1st 2017 of PSY4QV3R1 analysis. The small state differs among members, and is given for member M$XX$ by the small state of January 1st 20$XX$ of PSY4QV3R1 analysis. Large and small scale states are computed applying respectively 100-km low pass and high pass filters on PSY4QV3R1 outputs.

## 2.3 Indicators

We define here indicators quantifying the upwelling intensity as well as the impact of OIV and of tides and rivers. As explained by Da et al. (2019), upwelling indicators can be built based on surface wind (Ekman transport theory) or SST (see Benazzouz et al., 2014, for a review). SST-based indicators inform about the upwelling intensity, but also on the spatial distribution of upwelled water that triggers primary production at the surface. They can moreover be applied on satellite-derived SST data to study and monitor the upwelling.

### 2.3.1 Upwelling areas and intensity

Fig. 1 shows the June-September (hereafter JJAS) mean of the ensemble average STT of the FULL, NoTide and NoRiver ensemble. We highlight the four main areas of development of the SVU defined by To Duy et al. (2022) based on the spatial distribution of SST and upwelling intensity in the LONG simulation: BoxSC and BoxNC for respectively the southern (SCU)

and northern (NCU) coastal upwelling, BoxOF for the offshore upwelling (OFU), and BoxMK for the upwelling offshore the Mekong delta (MKU). Coordinates of the upwelling areas are provided in Fig. 1.

To quantify the intensity of upwelling, we use the indicators defined by Da et al. (2019), To Duy et al. (2022) and Herrmann et al. (2023). For any given point $(x, y)$ of the domain where $SST(x, y, t) < T_0$, the daily intensity is defined at day $t$ as:

$$UI_d(x, y, t) = T_{ref} - SST(x, y, t) \tag{1}$$

where $T_o = 27.6\,°C$ is the threshold temperature below which upwelling happens, $T_{ref} = 29.20\,°C$ is the reference temperature
over the area outside of the upwelling (see To Duy et al., 2022, for details). For a given upwelling area $B$ of size $A_B$, the daily upwelling intensity integrated over box $B$ is given at day $t$ by:

$$UI_{d,B}(t) = \frac{\iint_{(x,y)inB/SST(x,y,t)<T_o} (T_{ref} - SST(x, y, z, t))\, dx\, dy}{A_B} \tag{2}$$

Last, the intensity of upwelling integrated over the summer and the area $B$ is given by:

$$UI_{JJAS,B} = \frac{\int_{JJAS} UI_{d,B}(t)\, dt}{ND_{JJAS}} \tag{3}$$

where $ND_{JJAS} = 122$ is the number of days over JJAS.

Fig. 2 shows the area of upwelling development (highlighted by the 0.2°C isoline of the summer average of daily upwelling index $UI_d$) in 2017 and 2018 for the 10 members of the FULL ensemble and from 2009 to 2018 for the LONG simulation. Figs. 1 and 2 confirm that the boxes defined by To Duy et al. (2022) fully cover the upwelling development areas. MKU
does not develop along the coast, but $\sim$ 50 km off the Mekong delta coastline. It thus slightly covers the Mekong plume, as highlighted in Fig.2a by the difference between FULL and NoTide of JJAS sea surface salinity ensemble average.

### 2.3.2 Ocean Intrinsic Variability

To quantify the impact of OIV on daily and yearly upwelling indexes for a given ensemble, we use the OIV indicators defined by Herrmann et al. (2023).

$IV_{d,B}$ quantifies the effect of OIV on upwelling intensity over a given upwelling area at the daily scale:

$$IV_{d,B}(t) = \frac{\sigma_i(UI_{d,B}(t))}{\sqrt{m_i(\sigma_t(UI_{d,B}(t))^2)}} \tag{4}$$

where $m_i$ is the ensemble mean and $\sigma_t$ and $\sigma_i$ the temporal and ensemble standard deviation.

$IV_{JJAS,B}$ quantifies the effect of OIV on upwelling indensity at the summer scale:

$$IV_{JJAS,B} = \frac{\sigma_i(UI_{JJAS,B})}{m_i(UI_{JJAS,B})} \tag{5}$$

### 2.3.3 Tides and rivers effect

To quantify the effect of tides and rivers on the upwelling intensity and intrinsic variability, we use the indicators defined by Da et al. (2019). We compute the relative differences $\Delta m$ and $\Delta \sigma$ between the FULL reference ensemble and the NoRiver or NoTide sensitivity ensemble of respectively the mean (that quantifies the intensity) and standard deviation (that quantify its intrinsic variability) of the 10-member vector of yearly upwelling index :

$$\Delta m(UI_{JJAS,B}) = \frac{m_{i,SIM}(UI_{JJAS,B}) - m_{i,FULL}(UI_{JJAS,B})}{m_{i,FULL}(UI_{JJAS,B})} \times 100 \tag{6}$$

$$\Delta \sigma(UI_{JJAS,B}) = \frac{\sigma_{i,SIM}(UI_{JJAS,B}) - \sigma_{i,FULL}(UI_{JJAS,B})}{\sigma_{i,FULL}(UI_{JJAS,B})} \times 100 \tag{7}$$

where $SIM$ denotes the sensitivity simulation NoRiver or NoTide. The sole values of $\Delta m$ and $\Delta \sigma$, reported in Tab. 1, do not allow to quantitatively estimate if those differences, hence if the effect of tides or rivers, are statistically significant or not. For that, we compute the p-values $p_m$ and $p_\sigma$ associated respectively with the t test (significance of the mean difference) and F test (significance of the standard deviation difference). We apply those tests on the 10-member vectors of $UI_{JJAS,B}$ in the reference and sensitivity simulations and report them in Tab. 1. We also perform those tests at the daily scale on $UI_{d,B}$ to assess the significance between reference and sensitivity simulations of the time series of ensemble average and intrinsic variability of upwelling intensity shown in Fig. 3, highlighting in colors the periods when the differences are significant at more than 99% (p-value<0.01).

Last, we quantify the relationship between the daily chronology over JJAS of several variables (upwelling intensity over different areas as well as wind) by computing the Pearson correlation coefficient and associated p-value (that quantifies the statistical significance of the correlation) between the 122-day time series of those variables.

## 3 Impact of tides and rivers in the four upwelling areas

We examine for the four upwelling areas the effect of tides and rivers from a statistical point of view, based on the analysis of the daily and summer indicators introduced in Section 2.3. Figs.1e,f show the maps of the difference of the ensemble average of the mean JJAS SST in NoTide and NoRiver compared to FULL. Fig. 3 shows the daily time series of wind stress averaged over the entire upwelling region and of the ensemble average of $UI_{d,B}$ and of its intrinsic variability $IV_{d,B}$ for each upwelling area $B$ and each ensemble. Table 1 shows for each ensemble and each area the values of the summer integrated upwelling intensity $UI_{JJAS,B}$ for all members, of their ensemble average and standard deviation, and of the differences between the sensitivity ensembles NoRiver and NoTide and the reference ensemble FULL with their level of statistical significance.

## 3.1 Upwelling intraseasonal chronology

Herrmann et al. (2023) showed that the daily to intraseasonal variability of MKU, SCU and OFU is similar and driven at the first order by the variability of wind over the upwelling region. NCU does not develop during the core of the summer but in late June and late August (Fig. 3). To Duy et al. (2022) and Herrmann et al. (2023) indeed showed respectively for summers 2009 to 2018 and summer 2018 that NCU intraseasonal variability is driven first by the large-scale circulation. The SEJ and eddy dipole that form this large-scale circulation are well established in July-August, preventing NCU to develop. They are much weaker at the beginning (June) and end (September) of summer (or during summers of very weak wind), allowing NCU to develop. This development then depends on the organization of strongly chaotic small scale circulation and consequently shows a very high intrinsic variability.

In the NoTide and NoRiver ensembles, the daily to intraseasonal chronology of upwelling intensity is very similar, and highly statistically correlated (p-value < 0.01) to that of the FULL ensemble for each upwelling area (Fig. 3 and Table 1, column 16). Correlation exceeds 0.96 except for MKU for the NoTide ensemble (0.83, p < 0.01).

## 3.2 Influence of rivers

The effect of river discharge is significant, though weak, on MKU. Removing river discharges increases the summer ensemble average of $UI_{JJAS,boxMK}$ by 7% (the p-value associated to the differences between FULL and NoRiver $p_m = 0.02$, Table 1). This river effect is mainly significant during the transition period between two peaks of wind and upwelling, in particular before and after the strong mid-July and mid-August peaks (Fig. 3h): removing river discharge slightly but statistically significantly ($p_m < 0.01$) increases the ensemble average of MKU intensity $UI_{d,boxMK}$, by up to 30% on July 31st. The effect of river discharge on the intrinsic variability of $UI_{d,boxMK}$ is much weaker: $p_\sigma < 0.01$ only during three days, during which MKU intensity and intrinsic variability are anyway very weak (Figs. 3h,i).

The effect of river discharge on the ensemble average and intrinsic variability of OFU, NCU and SCU is not statistically significant neither at the summer integrated nor at the daily scale. $p_m$ and $p_\sigma$ are indeed larger than 0.1 for $UI_{JJAS,B}$ (Table 1, columns 12,15), JJAS SST differences are negligible (Fig.1e), and $p_m$ and $p_\sigma < 0.01$ are obtained for $UI_{d,B}$ only during very short periods (2-3 days maximum) and/or during periods of very weak upwelling (Figs. 3b-g).

## 3.3 Influence of tides

The most striking and significant effect highlighted by our sensitivity simulations is the influence of tides on MKU. This is first shown by the difference of summer averaged of the SST ensemble average: the difference between NoTide and FULL reaches 0.8°C over a large part of BoxMK (Fig.1d). The ensemble average value of $UI_{JJAS,boxMK}$ consequently decreases by 72%

($p_m < 0.01$) when tides are removed (Table 1). This effect of tides on the ensemble average of $UI_{d,boxMK}$ intensity is highly statistically significant during the whole summer (p-value < 0.01, Fig. 3g). Their effect on intrinsic variability is however much weaker, with $p_\sigma < 0.01$ only during a short period in June (Fig. 3i).

Tides do not significantly affect the ensemble average and intrinsic variability of SCU intensity at the summer integrated scale ($p_m$ and $p_{sigma} > 0.07$, Table 1). However, at the intraseasonal scale, removing tides induces a statistically significant decrease of the ensemble average of $UI_{d,boxSC}$ that can reach $\sim 20\%$ ($p_m < 0.01$) during the transition period of relatively low SCU intensity between the July and August wind peaks (Fig. 3d). The effect of tides on the intrinsic variability of $UI_{d,boxSC}$ is not statistically significant ($p_{sigma} > 0.01$, Fig. 3e).

Tides have a negligible influence on NCU and OFU at the summer integrated and daily scales. For those areas, the difference of JJAS SST between NoTide and FULL is negligible, hardly reaching 0.3°C (Fig.1d). $p_m$ and $p_\sigma$ for $UI_{JJAS,B}$ are larger than 0.1 (Table 1) and periods of $p_m$ and $p_\sigma < 0.01$ for $UI_{d,B}$ only occur when the upwelling intensity is very weak (Figs. 3d,e,f,g).

This analysis of spatially integrated indicators shows that river discharges and tides have a negligible influence on the daily to intraseasonal chronology in the four upwelling areas. For OFU, SCU and MKU, wind therefore remains the main driver of the upwelling chronology, whereas for NCU, the effect of large scale circulation predominates. Tides, and to a lesser extent river discharges, however significantly influence the intensity of MKU. Tides also influence, but to a second order extent, SCU intensity. Rivers do not significantly influence NCU, OFU and SCU, and tides have a negligible influence on NCU and OFU. The existence of MKU was recently revealed by To Duy et al. (2022), who suggested that, given its very low chaotic variability, MKU was induced by non chaotically varying processes, as topography or tides. In the following, we examine which physical mechanisms induce the development of MKU and explain the influence of river discharge and tides on MKU and SCU.

## 4 Physical mechanisms of MKU development

As already shown by Herrmann et al. (2023), the intrinsic variability of MKU is very low, with values of $VI_{tm}(UI_{JJAS,boxMK})$ smaller than 15% on average over the year (Table 1) and values of $VI_d(UI_{d,boxMK})$ not exceeding 40% at the daily scale for FULL and NoRiver and 60% for NoTide (Fig. 3i). In other words, the spreading of MKU intensity over the 10 members of a given ensemble is very small. To investigate the physical mechanisms involved in the development of MKU, we therefore examine and compare in this section one member of each ensemble FULL, NoTide and NoRiver. We take M17, i.e. the simulation where both the small and large scale states are given by conditions of January 2017 of the PSY4QV3R1 analysis (see Section 2.2). In this section, for the sake of simplicity, FULL, NoTide and NoRiver refer respectively to member M17 of the FULL, NoTide and NoRiver ensembles.

## 4.1 Circulation mechanisms

Figs. 4a,d,g shows for member M17 of the FULL ensemble the surface and bottom temperatures and their difference on 16/07/2018, the day of the July peak of MKU (Fig. 3h). Figs. 5a,d,g shows the surface and bottom horizontal velocity and the surface vertical velocity on the same day. The detailed bathymetry of the area is shown in Fig. 4i, as well as the location of points A to G, used in the analysis below.

In the Gulf of Thailand, a warm surface current (reaching 30°C) flows southward, following the 40 m isobath (Figs. 4a,5a). A branch of this Gulf of Thailand current veers east at 8°N around a seamount that rises to 20 m (north of point F, Fig. 4i). Over and north of the mount, the average surface current is extremely weak (less than 5 cm.s$^{-1}$, see the surface velocity maps as well as the profile of horizontal velocity at point F, Figs. 5a, 6f). This horizontal gradient of surface current velocity along the southern flank of the mount results in a divergence associated with positive values of vertical velocity reaching 2 to 5.10$^{-5}$ m.s$^{-1}$, i.e. 1 to 4 m.day$^{-1}$ (Fig. 5g). The induced upwelling partly explains the surface cooling observed in this area (SST < 28°C, Fig. 4a).

South off the Gulf of Thailand, a northward surface current flows with velocities exceeding 1 m.s$^{-1}$ (Fig. 5a) and relatively cold temperature compared to the surrounding surface waters ($\sim$ 28°C, Fig. 4a). It belongs to the large scale anticyclonic circulation that prevails in the southeastern SCS. This cold current meets a second branch of the warm southward Gulf of Thailand current, and they both bifurcate northeastward in the area 103.5-105°E / 4.5-7.5°N. This convergence results in a front of negative surface vertical velocity ($\sim -2.10^{-5}$ m.s$^{-1} \sim -2$. m.day$^{-1}$, Fig. 5g) thus in a downwelling of surface water. The downwelled water flows northeastward on the shelf bottom, with velocities reaching 30 cm.s$^{-1}$ (Fig. 5d) and a temperature of $\sim$ 28°C much warmer than the surrounding bottom water (< 26°C, Fig. 4d). This area of downwelled water flow corresponds to the area where the difference between the surface and the bottom temperature is negligible (< 1°C in Fig.4g), i.e. the water column is rather homogeneous. These results are confirmed by the vertically homogeneous profiles of temperature and northeastward horizontal velocity at points A ($\sim$ 27.5°C and $\sim$20 cm.s$^{-1}$, Fig. 6a) and B ($\sim$ 28.5°C and $\sim$30 cm.s$^{-1}$, Fig. 6b), located on the shelf downstream the area of surface convergence.

Further northeast, this barotropic warm northeastward current meets on the shelf slope a cold bottom northwestward current that flows from the open area towards the coast (Figs. 4d,5d). At the surface of this bottom convergence, a strong divergence of the northeastward alongshore surface current occurs, with values increasing abruptly from $\sim$ 30 cm.s$^{-1}$ to $\sim$ 60 cm.s$^{-1}$ (Fig. 5a). This can be seen more quantitatively on the profiles of temperature and velocity at points B and C, located respectively west (upstream) and east (downstream) of the front of bottom convergence and surface divergence (5g). Point C shows a cold (< 26°C) and northwestward current in the bottom layer below 20 m depth and a warm and fast northeastward current in the surface 0-20 m layer (28.5°C and $\sim$ 50 cm.s$^{-1}$ at the surface, Fig. 6c). This current is almost twice faster at the surface than the current at point B ($\sim$ 28.5°C and $\sim$ 30 cm.s$^{-1}$, Fig. 6b), which is northeastward over the entire depth. This bottom convergence of warm northeastward and cold northwestward currents and the divergence of the northeastward surface

current are associated with a front of positive vertical velocity between 106°E and 107°E and 6°N and 8.5°N, that reaches $\sim 6.10^{-5}$ m.s$^{-1} \sim 5$ m.day$^{-1}$ (Fig. 5g). The relatively cold bottom water ($< 27$°C) that results from the meeting of the warm and cold bottom water masses is upwelled and advected northward during this upwelling. It finally emerges at the surface near 8.5°N, where it is advected northeastward by the alongshore current (Figs. 4a,5a). Points D and E, located respectively further north and downstream the convergence region, indeed show homogeneous cold (respectively 27.5°C and $< 27$°C) profiles with northeastward velocities (Figs. 6d,e).

The upwelling that develops offshore the Mekong mouth can therefore be explained by the divergence/convergence associated with gradients of horizontal current resulting from the interactions over a marked topography of coastal and offshore surface and bottom circulation that prevails over the area. To further highlight the role of topography, we performed an additional simulation with the same configuration as member M17 of the FULL ensemble, but where we smoothed the topographic anomalies observed along the area of high upward vertical velocity observed in Fig. 5g (Figs. 7a,b). The 10 members of the FULL ensemble as well as the LONG simulation show extremely similar positions and values of strong surface velocity (see Figs. 7c-e that show the surface vertical velocity for those simulations on July 16th, 2018, the day of MKU intensity peak). In the simulation with smoothed bathymetry, the position of the area of strong vertical surface velocity is modified, shifted by $\sim 30$ km to the West (Fig. 7f). This sensitivity simulation therefore confirms the role of topography in determining the spatial extension of MKU.

## 4.2 Impact of tides

Fig. 8 shows the maps of tidal ellipses and currents for the three main tidal components over the MKU region: O1, K1 and M2. It also shows the 0.2°C iso-contour of difference between the JJAS SST in the ensemble averages of FULL and NoTide, corresponding to the area of stronger upwelling in FULL than in NoTide, southeast of point F and over the eastern slope of the Mekong shelf. Strong tidal currents with velocities reaching 30 cm.s$^{-1}$ for O1, K1 and M2 develop along the southeastern part of the Gulf of Thailand, in the surface convergence area described in Sec. 4.1 (around point F), over the shelf along the Mekong mouth, and over the shelf downstream the surface divergence area (near points D and E). These areas of strong tidal currents are contiguous to the region of strong JJAS SST difference between FULL and NoTide.

We show below that tides actually influence the development of MKU through two mechanisms: their influence on currents and the mixing of cold bottom water with warm waters from shallow areas. Fig. 5i shows the tidal surface currents on 16/07/2018 for member M17 of the FULL ensemble. Figs. 4b,e,h shows the surface and bottom temperatures and their difference and Figs. 5b,e,h shows the surface and bottom horizontal velocity and the surface vertical velocity for NoTide. Figs. 4c,f and Figs. 5c,f show the difference of bottom and surface temperature and horizontal velocity between FULL and NoTide.

### 4.2.1 Tidal influence on the current velocity and its horizontal gradient

The first effect of tides is their impact on the average current. In NoTide, an alongshore coastal current flows over the whole depth, southward in the Gulf of Thailand then northeastward along the Mekong mouth (Figs. 5b,e). In the FULL simulation with tide, this current is non-existent to very weak: its velocity is reduced by more than 75% (Figs. 5c,f). For points A, B, D, E, located along this current in NoTide, the velocity decreases from $\sim$40-50 cm.s$^{-1}$ in NoTide to $\sim$ 20 cm.s$^{-1}$ in FULL (Figs. 6a,b,d,e). For point F the difference is even stronger, with a 60 cm.s$^{-1}$ surface current in NoTide vs. no current in FULL. The area where the alongshore coastal current is weakened when tides are taken into account (i.e. where the reduction of velocity difference in FULL compared to NoTide exceeds 50%, Figs. 5c,f) coincides with the area of strong tidal currents, where the total tidal current reaches average daily velocity of 50-60 cm.s$^{-1}$ (Fig. 5i). As shown over the Yellow Sea (Moon et al., 2009; Wu et al., 2018; Lin et al., 2022), this weakening of current flowing over the shallow coastal area is due to the effect of tidal currents, that increase the bottom friction and background turbulence, due to the nonlinear interaction between wind-driven current and tides in the quadratic bottom friction term (Hunter, 1975).

Conversely, tide weakly modifies the strong large scale cyclonic current that originates from the southwest of the domain with speed reaching 1 m.s$^{-1}$ and flows northeastward following the shelf slope (Figs. 5a,b). It even slightly increases locally by about $\sim$20% (Fig. 5c): the surface velocity downstream the upwelling front increases from 40 cm.s$^{-1}$ in NoTide to 50 cm.s$^{-1}$ in FULL (Figs. 5a,b and point C, Fig. 6c). The horizontal velocity gradient between the regions of strong northeastward current offshore the Mekong shelf and of weak current over the shelf is consequently stronger in FULL than in NoTide, both at the surface and the bottom (Figs. 5a,b,d,e). Similarly, the gradient south of the mount near point F strongly increases in FULL. Though still existing, the surface and bottom convergence and divergence associated to the horizontal velocity gradient upstream and downstream the Mekong shelf are therefore weakened when tides are removed (Fig. 5h). This first effect of tides on the horizontal velocity gradient thus explains the weakening of the associated downwelling and upwelling in NoTide compared to FULL.

### 4.2.2 Effect of tidal mixing on vertical stratification

The second contribution of tides to the upwelling is related to the tidally induced vertical mixing of water masses. This tidal mixing contributes to the homogenisation of the water column and to the surface cooling, thus enhancing the effect of the horizontal gradient of surface and bottom currents described above. This effect occurs in specific areas of the domain, upstream and downstream the Mekong shelf. Fig. 9 shows for FULL and NoTide the profiles for points A to F of temperature and vertical diffusivity $k_z$, quantifying the vertical mixing. Point D, E and F are located in the area of strong tidal currents (Fig. 8), upstream the surface convergence zone (F, Fig. 5g) and downstream the surface divergence zone (D and E). For those points, $k_z$ is much stronger in FULL ($>10^{-2}$ m$^2$.s$^{-1}$) than in NoTide (between $10^{-5}$ and $10^{-2}$ m$^2$.s$^{-1}$, Fig. 9b). As a result, the temperature is vertically homogeneous in FULL, whereas NoTide shows a strong vertical temperature gradient of 1 to

2°C and warmer surface water and colder bottom water than FULL (Fig. 9a). Tidal mixing thus significantly contributes to the upwelling there. Conversely, for points A and B located on the Mekong shelf, the temperature is vertically homogeneous (though slightly warmer in NoTide compared to FULL by $\sim 0.5$°C) and $k_z$ is similar (between $10^{-2}$ and $10^{-1}$ m$^2$.s$^{-1}$) in both simulations. The low stratification in the shelf region between the surface convergence and divergence areas is thus not due to a
local effect of vertical tidal mixing over the shelf itself. It rather results from the advection by the northeastward current of the water downwelled and mixed by tides remotely, upstream the shelf. Beside vertical mixing, lateral mixing induced by bottom tidal currents explains the eastward intrusion of cold deep water over the western bottom shelf slope near 104°E (Fig. 4f). This cold water is then advected over the shelf by the alongshore current, thus also contributes to the water cooling in the downstream area.

This effect of tidal mixing allows MKU to be partially maintained during the mid-summer low wind period (late July - early August, Figs. 3a,h). During this period, MKU indeed reaches a local minimum in the FULL ensemble whereas it completely disappears in the NoTide ensemble. Fig. 6 shows the vertical profiles of temperature and horizontal currents for points A-F in FULL and NoTide on 31/07, i.e. the day of minimum wind and $UI_{d,boxMK}$ during this low wind period. Over the shelf, the
velocity of the northeastward surface current as well its gradient are reduced on 31/07 compared to 16/07, both in FULL and NoTide. In NoTide, velocity is about 40 cm.s$^{-1}$ at A, B and C on 16/07, and about 30 cm.s$^{-1}$ on 31/07, with negligible velocity difference between those points. The upwelling completely vanishes. In FULL, velocity on 31/07 goes from $\sim 20$ cm.s$^{-1}$ at A to $\sim 30$ cm.s$^{-1}$ at B and C, vs. respectively 20, 30 and 50 cm.s$^{-1}$ on 16/07. The difference between C and A thus decreases from $\sim 30$ cm.s$^{-1}$ on 16/07 to $\sim 10$ cm.s$^{-1}$ on 31/07. This strongly reduces the upwelling due to the horizontal velocity gra-
dient effect in FULL. Beside, points E and F (and to lower extent point D) still show fully homogeneous temperature profiles in FULL on 31/07, vs. stratified profiles in NoTide with respectively $\sim 0.5$ and 1°C temperature differences between surface and bottom. Tidal mixing therefore still occurs in FULL, inducing a partial surface cooling and contributing to maintain MKU. During the period of low wind, the contribution of the horizontal circulation gradient to MKU is therefore strongly weakened, and tidal mixing is a major factor in maintaining MKU.

Tidal mixing therefore enhances the upwelling both through local effect (direct mixing of water column) and remote effect (advection of water mixed upstream). This effect is eventually transmitted to SCU, through the advection of water upwelled in MKU to the SCU zone. This explains the significantly higher $UI_d$ for SCU during the mid-summer low wind period ($\sim 20\%$, Fig. 3b).

To quantify the respective contributions of local and remote effect in the surface cooling, we examine the maximum over the water column of the difference of vertical diffusivity coefficient $k_z$ between FULL and NoTide, $\Delta log_{10}(k_z)$. Fig. 9c shows the map of $\Delta log_{10}(Kz)$, together with the -0.4°C contours of the SST difference between FULL and NoTide, which highlights the area of surface cooling induced by tides, and the $T_o$ isotherm in FULL, which highlights the area of upwelling. A value
of $\Delta log_{10}(Kz)$ higher than 2 means that the vertical mixing induced locally by tides dominates the vertical mixing: upstream

the Mekong shelf (point F) and in the wake of Con Day island (points D, E, see profiles of temperature and $k_z$ in Fig. 9a,b). A value lower than 1 means that tides do not significantly contribute to the local vertical mixing: over the Mekong shelf between the surface convergence and divergence zones (point A downstream point F).

## 4.3 Effect of rivers

The NoRiver ensemble produces a sligthly higher MKU than FULL, with the strongest difference ($\sim$30%) of $UI_{d,boxMK}$ ensemble average on 31/07 during the mid-summer low wind period (Fig. 3h) and a 9% increase of $UI_{JJAS,boxMK}$ (Table 1). Figs. 10a,b shows the maps of the difference of sea surface salinity (SSS) and temperature (SST) between NoRiver and FULL on 31/07. Removing Mekong river discharge logically results in a strong increase of sea surface salinity near the Mekong mouth in NoRiver compared to FULL. The area of $\sim$ 1 psu SSS difference highlights the Mekong plume, which is removed
in NoRiver. The upwelling intensification in NoRiver occurs in the northeastern part of this plume, with a $\sim$ 1°C SST cooling compared to FULL (point G in Fig. 10b).

This removal of the upper layer of fresher hence lighter water in NoRiver induces a weakening of the vertical stratification. Figs. 10c,d show the vertical profiles of salinity and temperature for points A-G on 31/07 in both simulations. For points A
to F, located southern of the Mekong plume, temperature and salinity vertical profiles are very similar in NoRiver and FULL, even if slightly shifted. More specifically, the vertical gradients of temperature and salinity are similar in both simulations: differences between NoRiver and FULL of the surface-bottom difference are smaller than 0.1°C and 0.1 psu. For those points, the stratification is therefore not significantly modified by river discharge. Conversely, for point G, located in the MKU area downstream point E, but also in the Mekong plume area (Fig. 10a), those profiles are very different in NoRiver and FULL. The
salinity difference between the surface and the bottom is $\sim$ 1 psu in FULL, vs. less than 0.1 psu in NoRiver, and the temperature difference is 0.6°C in FULL, vs. 0.1°C in NoRiver. The stratification at point G is therefore significantly weakened in NoRiver. This stratification weakening makes the water column easier to mix vertically, facilitating the tidal vertical mixing, which is the main contributor to MKU in this area and during the transition period, as explained above. The resulting surface cooling finally explains the increase of MKU intensity in NoRiver. Rivers thus locally hinder the development of MKU through their
strengthening effect on the stratification.

This area of Mekong plume influence on the stratification and on MKU is however small, as highlighted by Figs. 10a,c (limited to point G) and confirmed by Fig. 2: in the area where upwelling develops, the sea surface salinity difference between FULL and NoRiver does not exceed 1.0 psu. The area where strong buoyancy gradients could develop and enhance chaotic
variability therefore does not reach MKU region, explaining the non significant effect of river discharge on MKU OIV (Fig. 3i). The JJAS SST difference between FULL and NoRiver moreover shows that the river effect on the stratification is not sufficient to significantly enhance the Ekman transport driven upwelling at the coast (Fig. 1f).

## 5 Representativeness of summer 2018

Wind and upwelling intensity over the SVU region were stronger than average during summer 2018, and weaker during summer 2017 (To Duy et al., 2022). To discuss the representativeness of the results and conclusions obtained from the analysis of summer 2018, we thus examine the spatial and temporal variability of upwelling for summer 2017 in the FULL, NoTide and NoRiver 2-year ensembles. Fig. 11 shows for 2017 the maps of summer SST in the three ensembles and Fig. 12 shows the daily time series of wind and upwelling indicators.

Analysis of summer 2017 is in agreement with the analysis of summer 2018. First, the intraseasonal chronology of OFU, SCU and MKU intensity is primarily driven by wind (Fig. 12a,b,d,h, with highly significant correlations between 0.55 and 0.68, p-value <0.01 between $UI_{d,B}$ and wind for both summers). It is not the case for the intraseasonal chronology of NCU (Fig. 12a,f), that only develops at the beginning and end of summer 2017, confirming our conclusion about the blocking role of the core of summer general circulation. Second, the influence of OIV on upwelling intensity is very weak for MKU, weak for SCU, and stronger for OFU and NCU (Fig. 12c,e,g,i). Third, tides have a major role in MKU development both for 2017 and 2018, with no MKU developing at all in the NoTide ensemble for summer 2017 (Fig. 11e and Fig. 12h). Rivers slightly reduce MKU intensity in the middle of summer, whereas neither tides nor rivers significantly impact the upwelling over the three other areas (Fig. 11e,f and Fig. 12b-h).

To complete this analysis, we examine the summer upwelling location for different years/members in Fig. 2. For the 10 members of the FULL ensemble, MKU shows rigorously the same location of development for summer 2018, and similarly for summer 2017 though its extension is strongly reduced. Between 2009 and 2018 in the LONG simulation, MKU also always develops over the same core area, with a spatial extension varying with the strength of the upwelling. Those results, together with the sensitivity analysis to topography (section 4.1), confirm the stability of MKU location, related to the influence of topography.

## 6 Conclusions

We studied here the effect of tides and rivers on the intensity of SVU over its four areas of development at the daily to intraseasonal scale, and explored the detailed mechanisms explaining the development of upwelling over the Mekong shelf, MKU. For that, a reference and two sensitivity ensembles of 10 members were performed for the case study of the strong upwelling that occurred during summer 2018 along and offshore the Vietnamese coast.

Tides contribute strongly to MKU intensity, increasing its summer average by a factor 3.5, whereas rivers sligthly reduce it, by 9% on the summer average. Tides slightly increase SCU intensity during the mid-summer low wind period, and rivers do not affect it. Tides and rivers do not have a statistically significant influence on the upwelling in the offshore and northern

coastal regions, OFU and NCU. For the fours upwelling areas, neither tides nor rivers significantly influence the upwelling intraseasonal chronology. For OFU, SCU and MKU, wind therefore remains the main driver of the upwelling chronology, whereas for NCU, the effect of large scale circulation predominates.

Circulation and tides interacting over a marked topography are the main factors that explain MKU development. The interactions of a weak coastal alongshore shallow current flowing over the Mekong shelf and strong surface and bottom currents that prevail offshore around the shelf locally induce strong convergence or divergence of horizontal currents on the western and eastern flanks of the shelf slope. This results in high vertical velocities (of the order of a few m.day$^1$), with a downwelling upstream (west) of the shelf and an upwelling downstream (east) of the shelf. Tidal currents first enhance the bottom friction and background turbulence, which weakens the coastal alongshore shallow current. This increases the horizontal velocity gradient and the resulting divergence and convergence and associated vertical velocities, hence intensifies the induced downwelling and upwelling. Second, tides contribute to the vertical and lateral mixing of cold bottom waters with shallower warmer water. The advection into the upwelling region of the resulting cold water masses further intensifies the surface cooling. This effect of tidal mixing explains most of the MKU maintaining during the low wind mid-summer period, during which the circulation mechanisms hardly induce the upwelling. The water upwelled in the MKU area is moreover advected into the SCU area, explaining the $\sim 20\%$ contribution of tides to SCU intensity during this period. Last, surface freshwater from the Mekong river slightly weakens the MKU intensity (by up to 30% during the low wind period) by strengthening the vertical stratification. MKU is therefore mostly driven by non chaotic processes (large scale circulation, topography and tides), which explains its negligible OIV compared to NCU and OFU. A further analysis of summers 2009 to 2017 suggests that those conclusions are robust throughout the different years and associated atmospheric, oceanic and river conditions. Fig. 13 summarizes those findings on the processes involved the functioning of MKU.

Following and complementing the previous studies about the SVU, this study allowed to deepen our understanding of the physical mechanisms involved in the functioning and variability of this upwelling over its different areas of development: wind, (sub)mesoscale dynamics, ocean intrinsic variability, tides, rivers, topography. New studies are now required to better understand the influence of the upwelling on the planktonic ecosystems, notably based on the use of coupled physical-biogeochemical modelling (e.g. SYMPHONIE-Eco3MS, Ulses et al., 2016; Herrmann et al., 2017). The surface cooling associated with the SVU and its different mechanisms is moreover part of the heat budget of the regional climate system. Several studies also highlighted the impact of the SVU on local atmospheric dynamics, including winds (Zheng et al., 2016; Yu et al., 2020). Using online closed budget tools, as done by Trinh et al. (2023) over the whole South China Sea, as well as ocean-atmosphere coupled, as the SYMPHONIE-RegCM recently developed by (Desmet, 2024), will now allow to better understand and quantify the role of the upwelling as well as other factors in those budgets, its feedback on the atmospheric dynamics and local climate, and its response to climate change, especially since climate projections predict a weakening of the summer winds (Herrmann et al., 2020, 2021). Applying perturbations on the atmospheric fields that drive the upwelling (i.e. the wind, see for example Nguyen-Duy et al., 2023) or to the lateral boundary conditions that would create a different submesoscale to mesoscale circula-

tion (i.e. the wind, see for example Da et al., 2019) would also introduce chaoticity in our simulations, and would be interesting to study. Beside those modeling efforts, dedicated in-situ campaigns would be extremely useful to confirm our conclusions, in particular for MKU for which extremely little data are available.

*Code and data availability.* The SYMPHONIE model is available on the webpage of the SIROCCO group, https://sirocco.obs-mip.fr/. Daily
sea surface temperature simulated by the FULL, NoTide and NoRiver ensembles over summer 2018, and tridimensional temperature, salinity and currents simulated by member M17 of the FULL, NoTide and NoRiver ensembles for 16/07/2018 and 31/07/2018 are freely available on https://doi.org/10.5281/zenodo.10626111.

*Author contributions.* Marine Herrmann and To Duy Thai designed the experiments and To Duy Thai carried them out. Patrick Marsaleix develops the SYMPHONIE numerical model. Marine Herrmann wrote the manuscript with contributions from To Duy Thai and Patrick
Marsaleix.

*Competing interests.* The authors declare that they have no conflict of interest.

*Acknowledgements.* This work is a part of LOTUS international joint laboratory (lotus.usth.edu.vn). PhD studies of To Duy Thai were funded through an IRD ARTS grant and a "Bourse d'Excellence" from the French Embassy in Vietnam. Numerical simulations were performed using CALMIP HPC facilities (projects P13120 and P20055) and the cluster OCCIGEN from the CINES group (project DARI A0080110098).

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

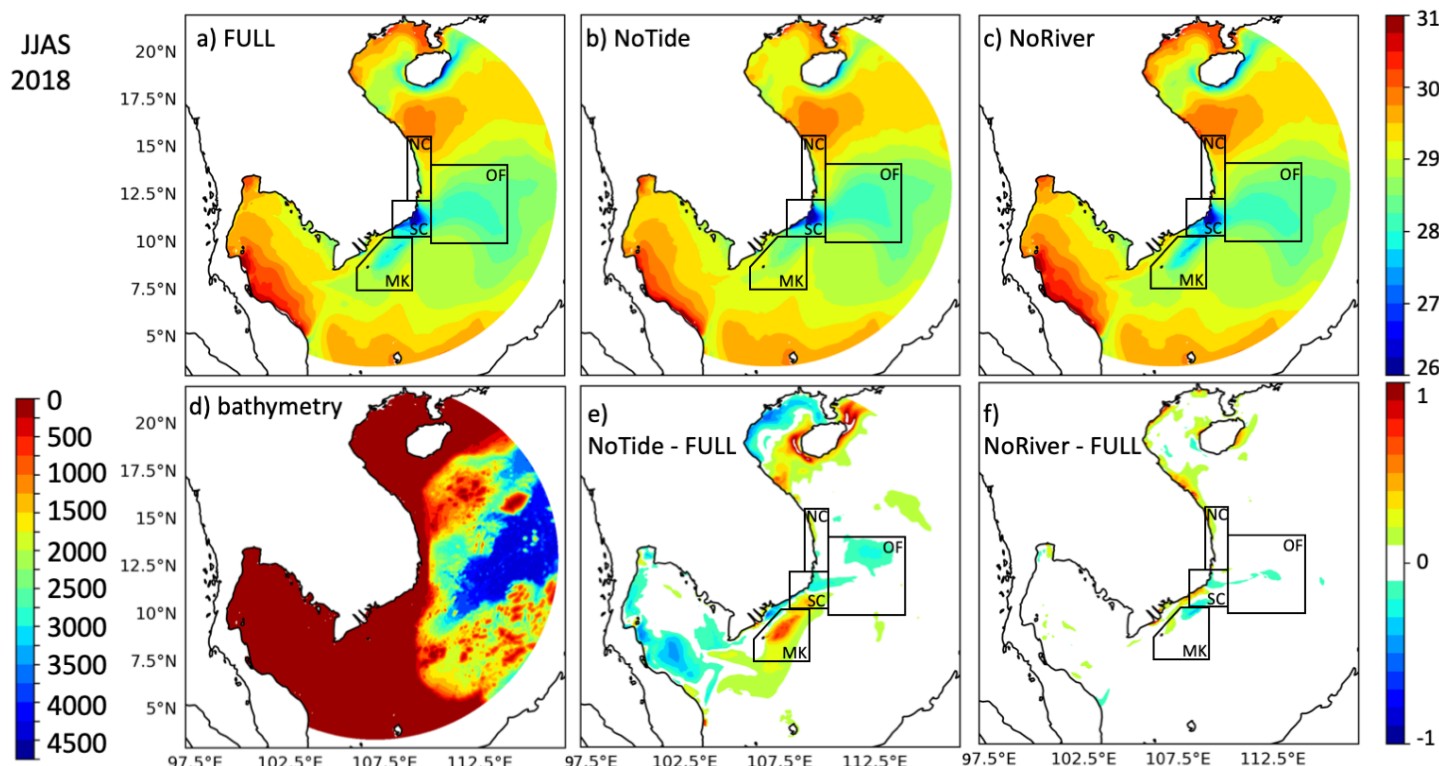

**Figure 1.** Ensemble average summer SST over June-September 2018 in the FULL (a), NoTide (b) and NoRiver (c) ensembles and difference between the NoTide (d) and NoRiver (f) and FULL ensembles (°C). (d) Bathymetry of the domain (m). Color bars for panels (a-c) and (e-f) are provided on the top and bottom right, and color bar for panel (d) on the bottom left. Coordinates of upwelling areas: BoxNC (12.2-15.5°N; 108.7-109.9°E), BoxSC (10.3-12.2°N; 108-109.9°E), BoxOF (10-14°N; 109.9-114°E), BoxMK (at depth > 17 m, 7.5-10.3°N; 106-109°E).

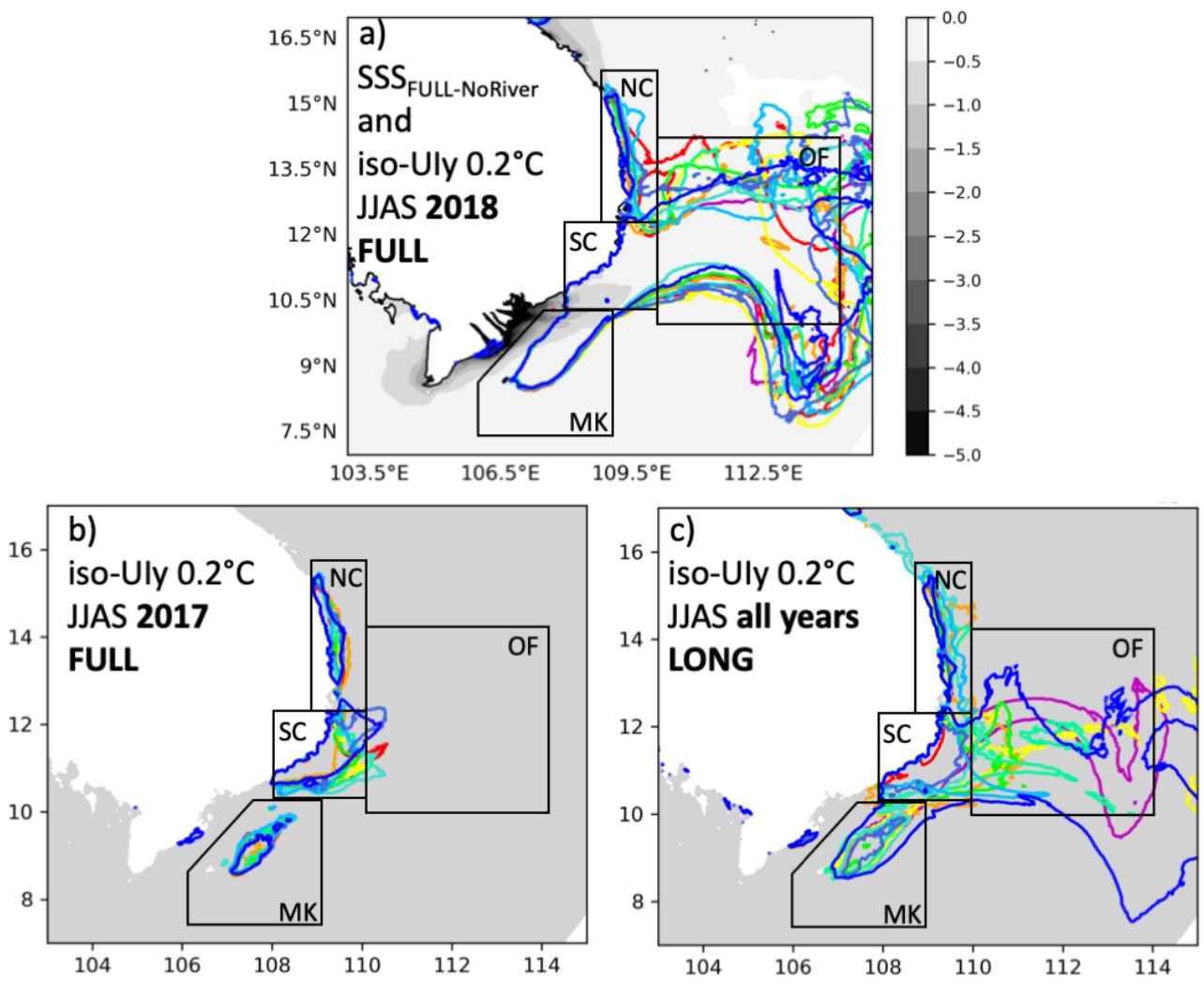

**Figure 2.** Ensemble average sea surface salinity over June-September 2018 in FULL (colors, a) and 0.2°C isoline of the summer averaged upwelling index for the 10 members of the FULL simulation in 2018 (a) and 2017 (b) and for the 10 years of the LONG simulation.

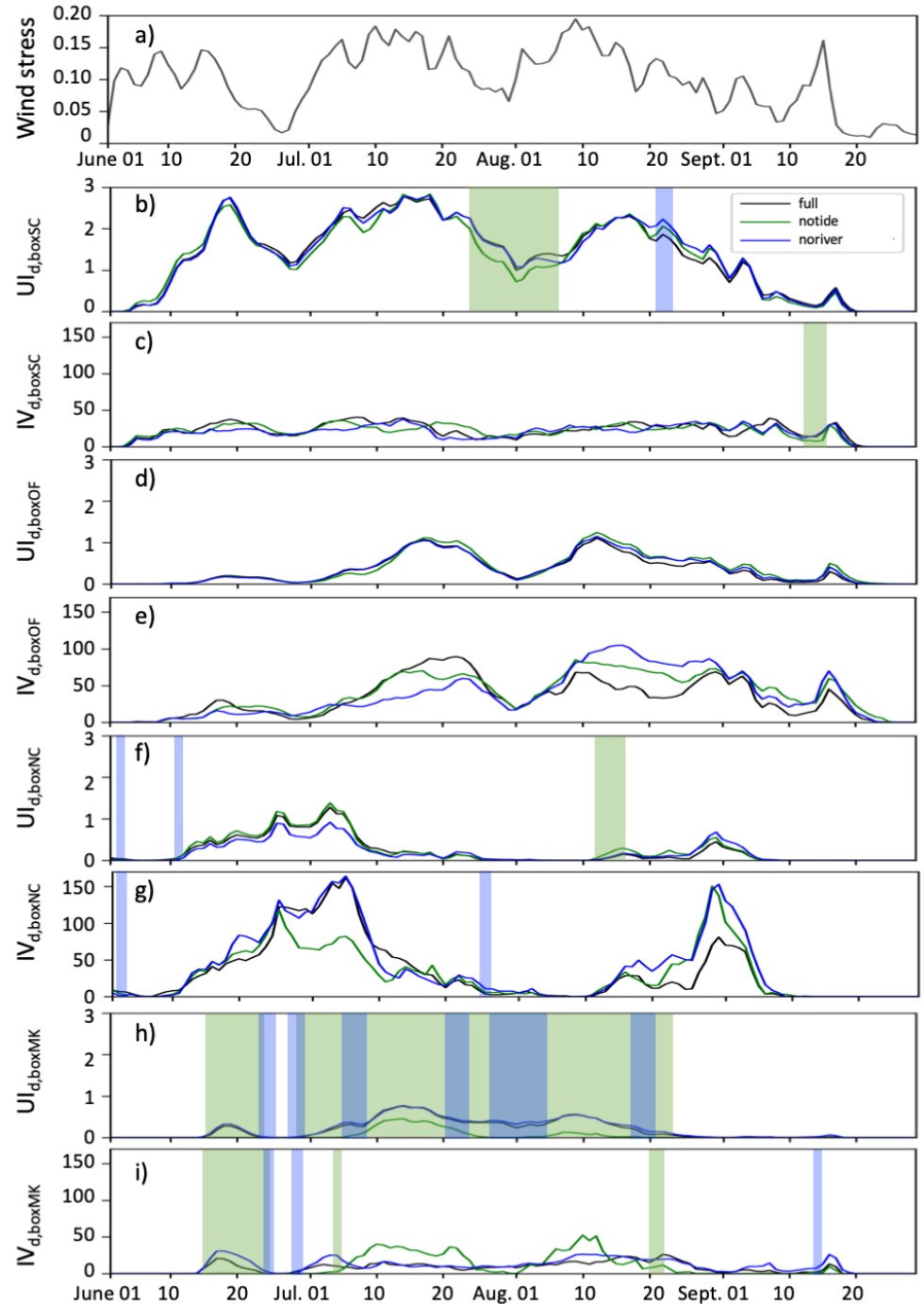

**Figure 3.** Daily time series over summer 2018 of averaged wind stress (a, N.m$^{-2}$) over the whole upwelling region (7.5-14°N, 106-114°E) and of the ensemble mean of $UI_{d,B}$ and of $IV_{d,B}$ (°C) for the FULL (black), NoTide (green) and NoRiver (blue) ensembles for NCU (b,c), SCU (d,e), OFU (f,g) and MKU (h,i). Shaded green and blue colors shows the areas where the difference between the reference FULL and sensitivity NoTide and NoRiver ensembles is statistically significant at more than 99%.

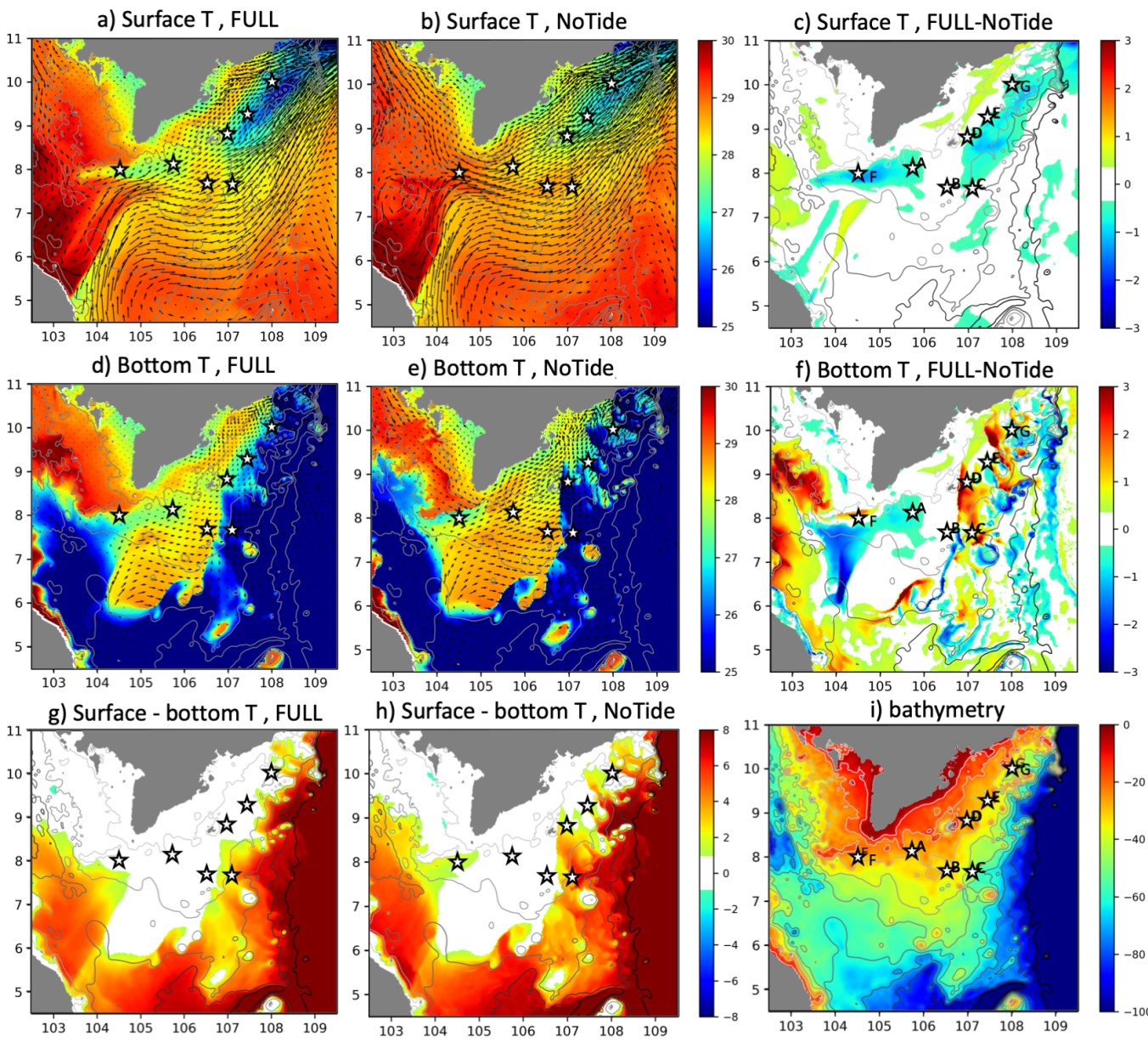

**Figure 4.** Left and middle columns: surface (a,b) and bottom (d,e) temperature and their difference (g,h) on 16/07/2018 in M17 of the FULL (a,d,g) and NoTide (b,e,h) ensembles (°C). Arrows show the surface (a,b) or bottom (d,e) horizontal velocity (see Fig. 5 for the values). Right column: (c,f): difference between FULL and NoTide (°C), (i): bathymetry (m) of the area, with isobaths from 10 to 100 m every 10 m.

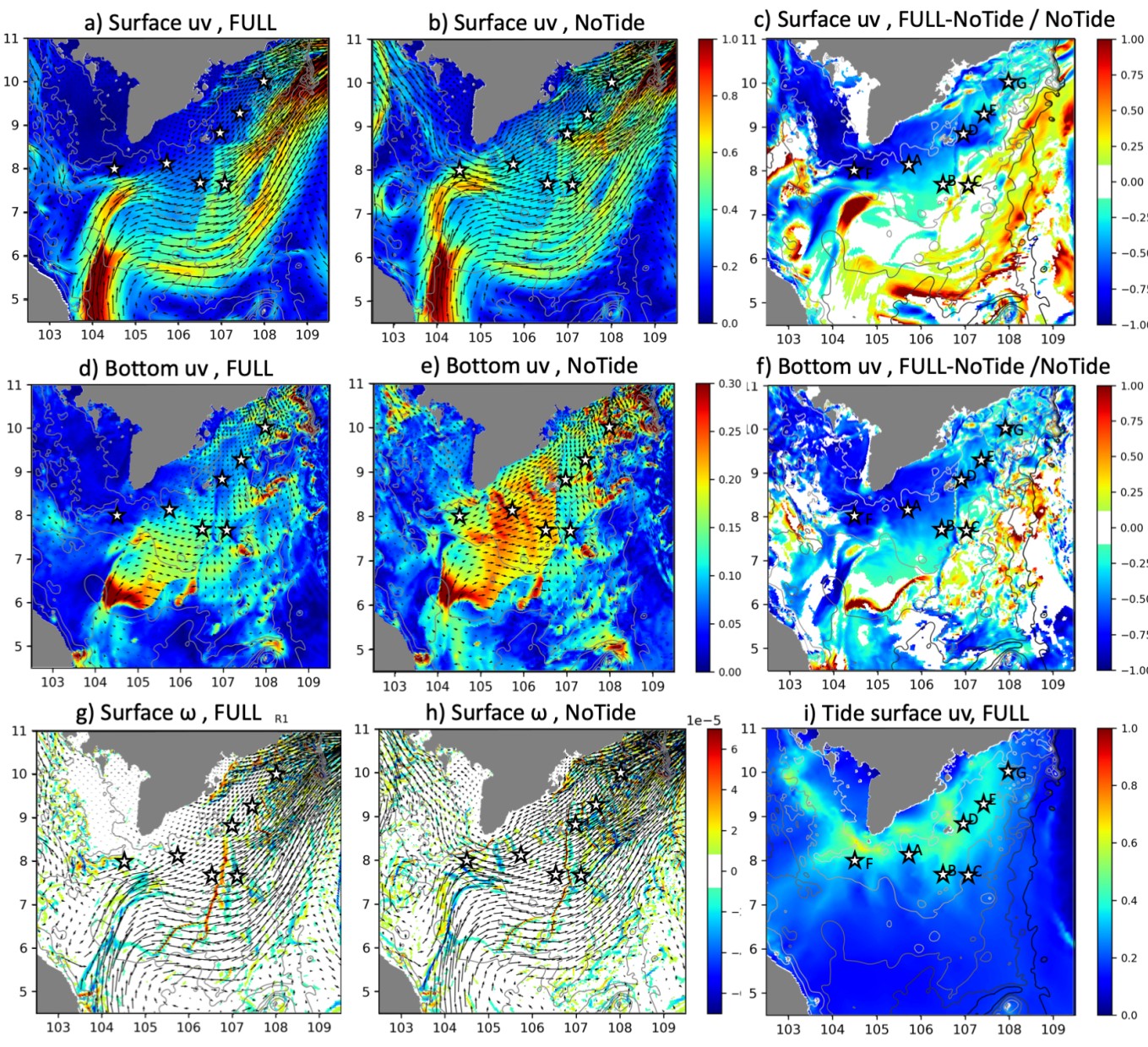

**Figure 5.** (a,b,d,e): surface (a,b) and bottom (d,e) horizontal velocity on 16/07/2018 in M17 of the FULL (a,d) and NoTide (b,e) ensembles (m.s$^{-1}$, colors indicate the speed, and arrows the direction). (c,f): relative difference between FULL and NoTide (no unit). (g,h): Surface vertical velocity in FULL (g) and NoTide (h) (m.s$^{-1}$, arrows show the surface horizontal velocity). (i): surface tidal speed in FULL (m.s$^{-1}$).

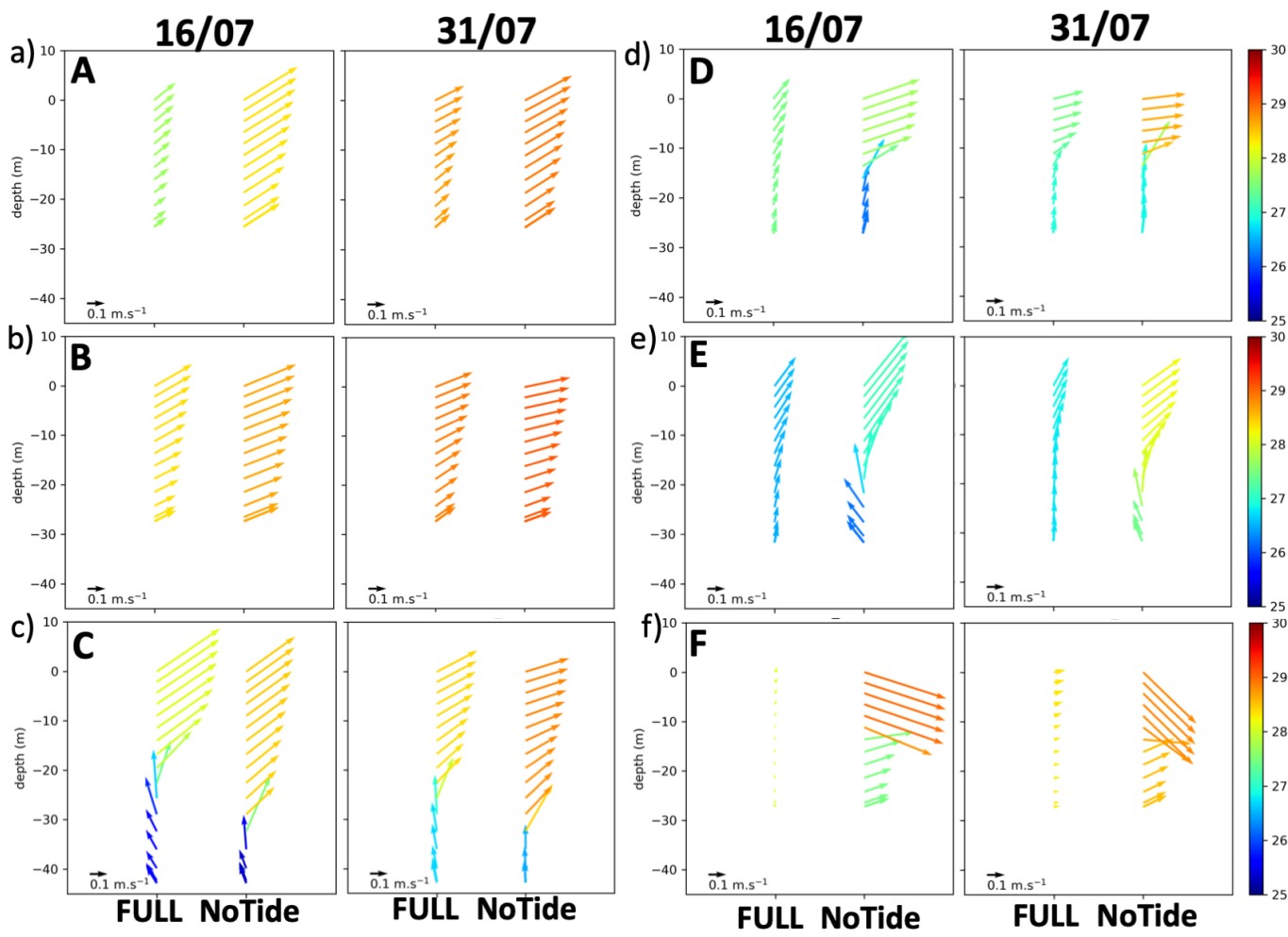

**Figure 6.** Profiles of temperature (color, °C) and horizontal velocity (arrows, m.s$^{-1}$) on 16/07/2018 and 31/07/2018 at points A, B, C, D, E,F shown in Figs. 4,5,8 in M17 of the FULL and NoTide ensembles.

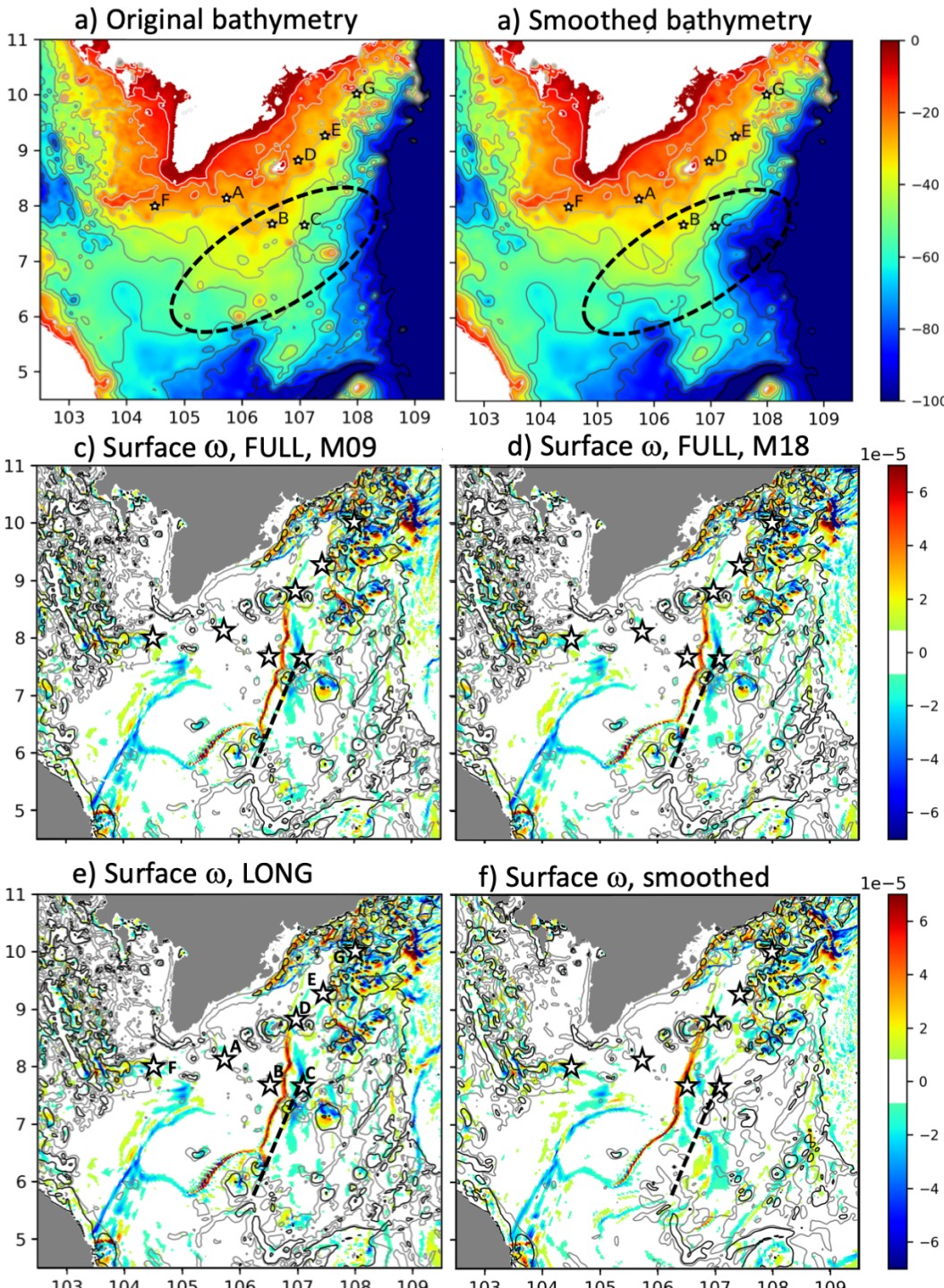

**Figure 7.** Initial bathymetry (a, m) and smoothed bathymetry (b), the dashed ellipse shows the area of bathymetry smoothing. Surface vertical velocity (m.s$^{-1}$) on 16/07/2018 in M09 (c) and M18 (d) of the FULL ensemble (the other members, not shown, are extremely similar), in LONG (e) and in the sensitivity simulation with smoothed bathymetry (f). The dashed line highlights the front of strong upward vertical velocity.

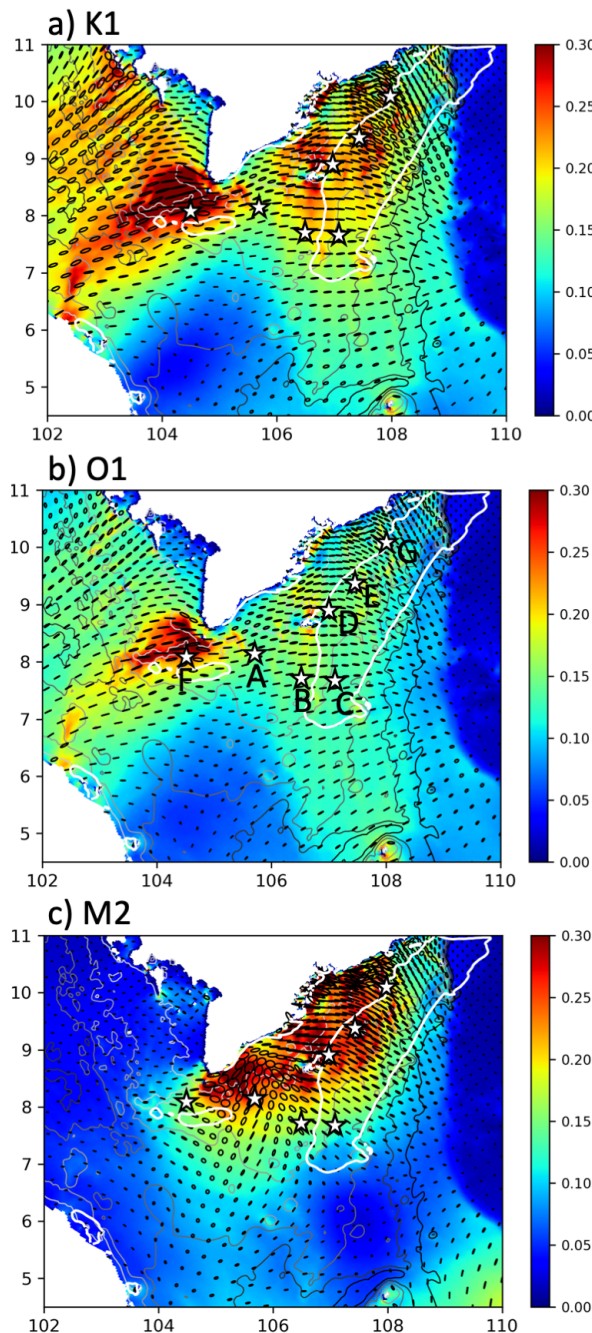

**Figure 8.** Depth-averaged tidal current ellipses and current intensity (m.s$^{-1}$) in M17 of the FULL ensemble for the 3 significant tidal components in the MKU region: K1, O1 and M2. The white lines show the 0.2°C isoline of the difference between JJAS averaged SST in FULL and NoTide, corresponding to the area where upwelling is much stronger in FULL than in NoTide in Fig. 1.

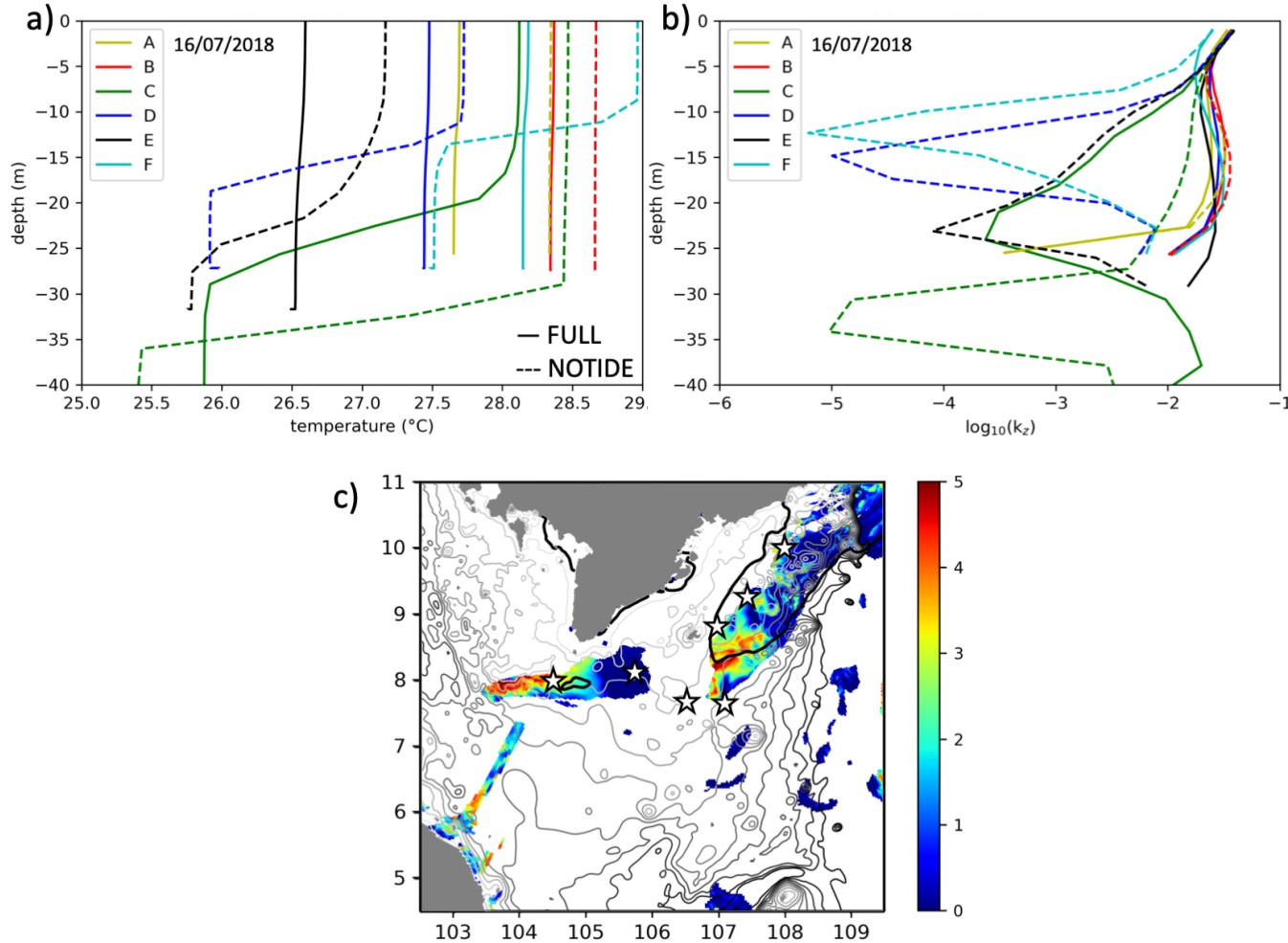

**Figure 9.** Profiles of temperature (a, °C) and diffusivity coefficient $k_z$ (b, m$^2$.s$^{-1}$, logarithmic scale) on 16/07/2018 at points A, B, C, D, E, F shown in Figs. 4,5,8 in M17 of the FULL (full lines) and NoTide (dashed line) ensembles. (c) Map on the same day of the maximum over the water column of the difference of diffusivity coefficient between FULL and NoTide, $\Delta log_{10}(k_z)$, plotted for areas where the FULL-NoTide SST difference exceeds 0.4°C. The black line shows the $T_o = 27.6$°C surface isotherm.

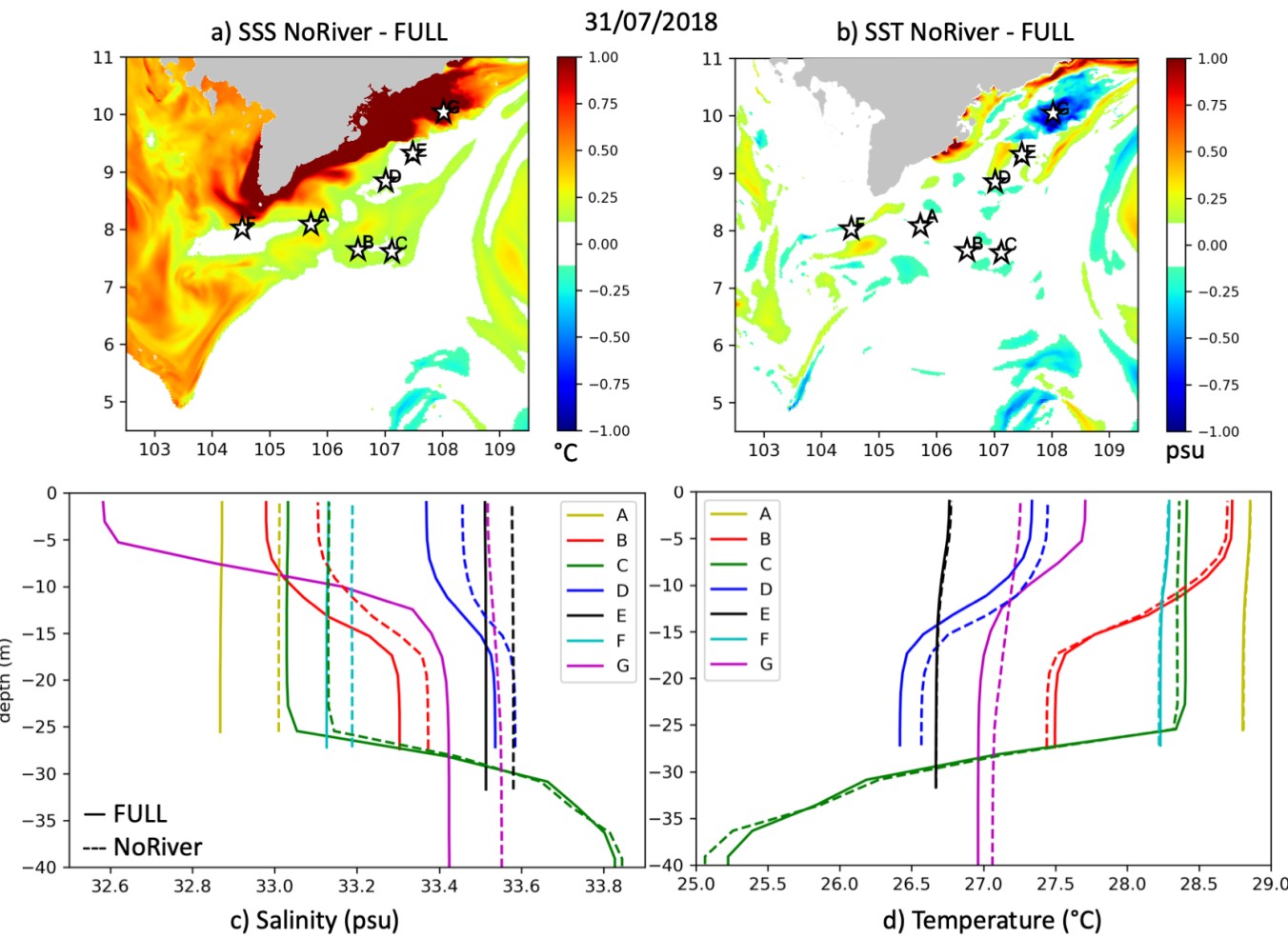

**Figure 10.** Top row: difference of sea surface salinity (a, psu) and temperature (b, °C) between M17 of the NoRiver and FULL ensembles over the MKU region on 31/07/2018. Bottom row: corresponding profiles of salinity (c) and temperature (d) over points A-G shown on panels (a-b) for FULL (full lines) and NoTide (dashed line).

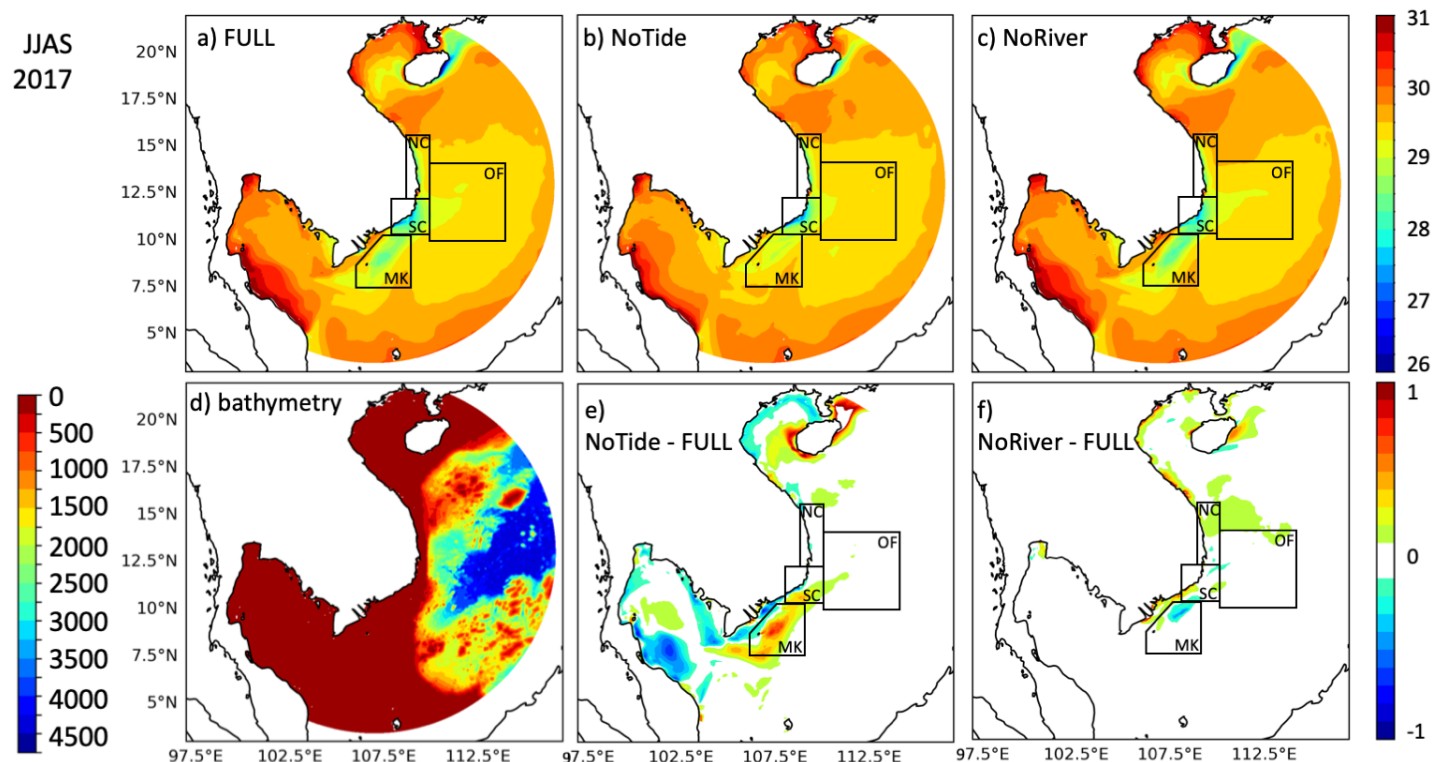

**Figure 11.** Same as Fig. 1 for summer 2017.

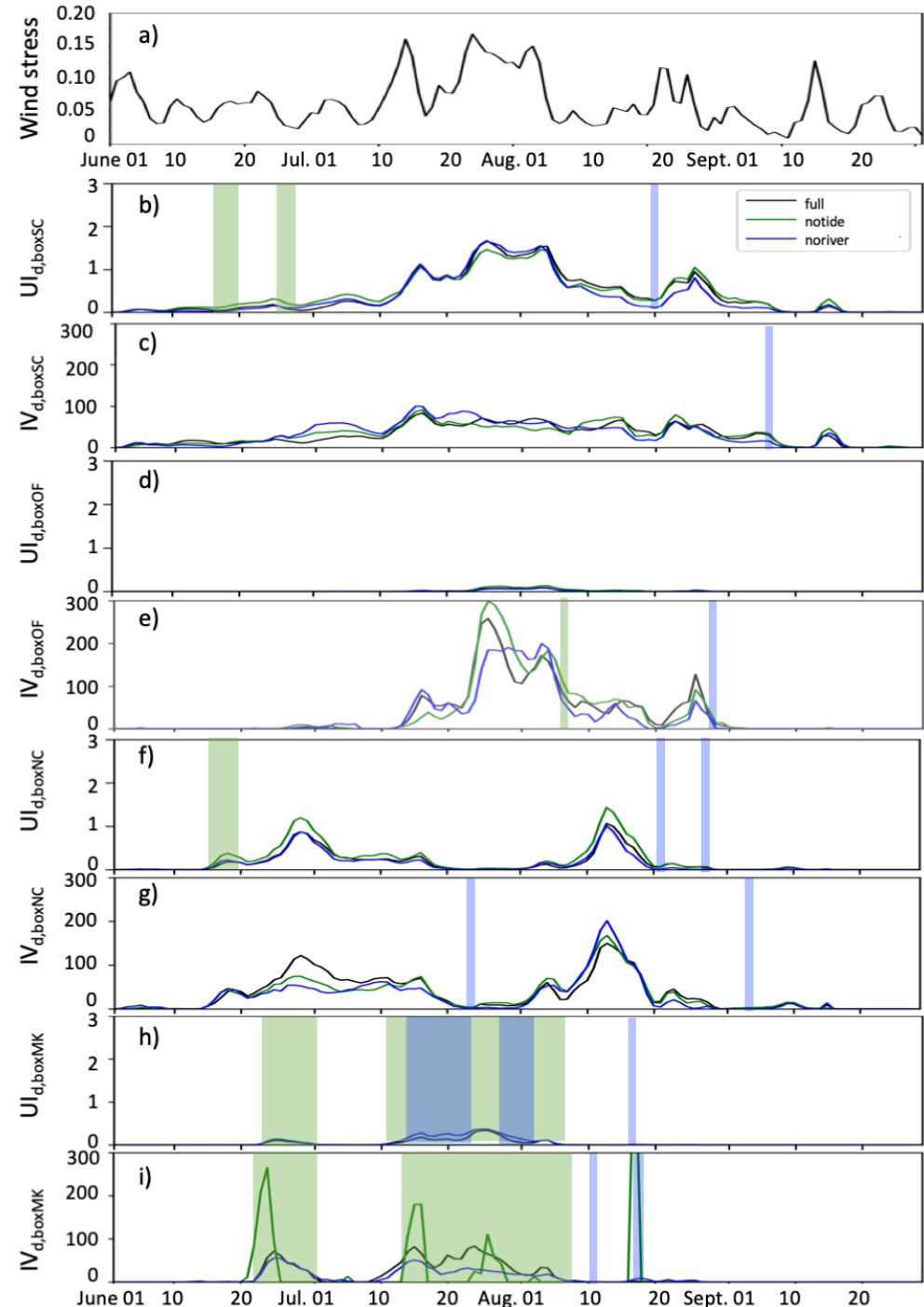

**Figure 12.** Same as Fig. 3 for summer 2017.

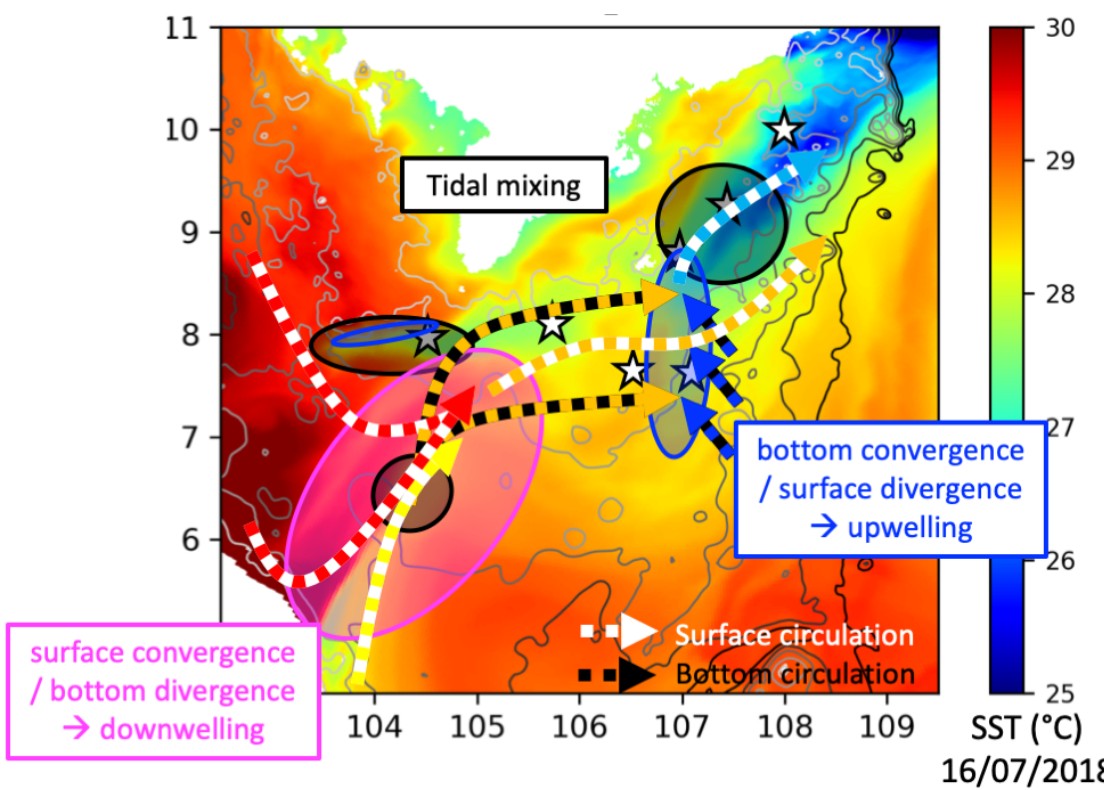

**Figure 13.** Schematic representation of MKU functioning. Arrows represent the surface (white) and bottom (black) circulation and the associated temperature (colors).

| Member | (1) M09 | (2) M10 | (3) M11 | (4) M12 | (5) M13 | (6) M14 | (7) M15 | (8) M16 | (9) M17 | (10) M18 | (11) $m_i$ | (12) $\Delta m(UI)$ ($p_m$) | (13) $\sigma_i(UI)$ | (14) $IV_{JJAS,B}$ | (15) $\Delta\sigma(UI)$ ($p_\sigma$) | (16) $c(UI_{d,b})$ vs. FULL |
|---|---|---|---|---|---|---|---|---|---|---|---|---|---|---|---|---|
| **OFU** | | | | | | | | | | | | | | | | |
| FULL | 0.32 | 0.35 | 0.26 | 0.39 | 0.37 | 0.39 | 0.26 | 0.26 | 0.42 | 0.31 | 0.33 | — | 0.059 | 18% | — | — |
| NoTide | 0.40 | 0.38 | 0.35 | 0.36 | 0.33 | 0.39 | 0.47 | 0.29 | 0.55 | 0.20 | 0.37 | 12.5% (0.26) | 0.095 | 26% | 59.9% (0.18) | 0.98 |
| NoRiver | 0.40 | 0.32 | 0.45 | 0.35 | 0.41 | 0.46 | 0.34 | 0.22 | 0.31 | 0.24 | 0.35 | 6.2% (0.52) | 0.080 | 23% | 35.2% (0.38) | 0.99 |
| **NCU** | | | | | | | | | | | | | | | | |
| FULL | 0.13 | 0.28 | 0.09 | 0.18 | 0.20 | 0.17 | 0.29 | 0.30 | 0.31 | 0.16 | 0.21 | — | 0.078 | 37% | — | — |
| NoTide | 0.24 | 0.21 | 0.30 | 0.18 | 0.18 | 0.31 | 0.20 | 0.28 | 0.32 | 0.17 | 0.24 | 12.8% (0.39) | 0.059 | 25% | -24.5% (0.41) | 0.99 |
| NoRiver | 0.13 | 0.26 | 0.21 | 0.16 | 0.34 | 0.28 | 0.14 | 0.09 | 0.12 | 0.11 | 0.18 | -13.2% (0.45) | 0.085 | 46% | 9.4% (0.79) | 0.96 |
| **SCU** | | | | | | | | | | | | | | | | |
| FULL | 1.25 | 1.35 | 1.25 | 1.49 | 1.39 | 1.42 | 1.27 | 1.38 | 1.48 | 1.29 | 1.36 | — | 0.091 | 7% | — | — |
| NoTide | 1.35 | 1.28 | 1.29 | 1.26 | 1.24 | 1.30 | 1.35 | 1.29 | 1.40 | 1.29 | 1.31 | -3.8% (0.13) | 0.047 | 4% | -47.6% (0.07) | 0.98 |
| NoRiver | 1.40 | 1.33 | 1.37 | 1.42 | 1.46 | 1.40 | 1.35 | 1.24 | 1.38 | 1.25 | 1.36 | 0.3% (0.92) | 0.070 | 5% | -23.1% (0.45) | 0.99 |
| **MKU** | | | | | | | | | | | | | | | | |
| FULL | 0.18 | 0.18 | 0.19 | 0.19 | 0.17 | 0.18 | 0.17 | 0.18 | 0.20 | 0.17 | 0.18 | — | 0.010 | 6% | — | — |
| NoTide | 0.05 | 0.04 | 0.05 | 0.05 | 0.05 | 0.06 | 0.05 | 0.04 | 0.07 | 0.05 | 0.05 | -71.7% (<0.01) | 0.007 | 13% | -32.0% (0.27) | 0.83 |
| NoRiver | 0.22 | 0.18 | 0.17 | 0.20 | 0.20 | 0.18 | 0.20 | 0.20 | 0.22 | 0.19 | 0.20 | 8.6% (0.02) | 0.016 | 8% | 57.73% (0.19) | 0.99 |

**Table 1.** Value of $UI_{JJAS,B}$ for each member of each ensemble (columns 1 to 11), ensemble mean (11), standard deviation (13) and their ratio $IV_{JJAS,B}$ (14), difference between the sensitivity and reference simulations of the ensemble mean (12) and the ensemble spread (15). Last column (16): correlation $c$ between $UI_{d,B}$ in the sensitivity and reference simulations. In colomns (12) to (15), $UI$ stands for $UI_{JJAS,b}$.