# Peer review of "Mechanisms and intraseasonal variability of the South Vietnam Upwelling, South China Sea: role of circulation, tides and rivers"

_EGUsphere, 2024_

## Author Comment (AC1)

Reviewer 1 Javier Zavala-Garay

We warmly thank Javier Zavala-Garay for the time and attention devoted to our paper, and for those positive and constructive comments. We have carefully considered all those comments and suggestions in the revised version of our manuscript. In what follows, our answers and modifications are highlighted in blue. Page and line numbers refer to the highlighted version of the revised manuscript.

**General comments**

This manuscript evaluates the role of the background circulation, tides, and rivers on the South Vietnam Upwelling (SVU) system during the summer of 2018. The methodology and analysis is well though and convincing evidence is presented supporting that:

1. Wind forcing is the leading mechanism explaining the chronology of the upwelling in the 4 areas studied

2. The upwelling during the summer of 2018 in the Mekong shelf (MKU) is due to the convergence and divergence of currents (and the associated downwelling and upwelling), which are strongly influenced by topography and to a lesser extent by tides.

3. Tides increase the horizontal velocity gradient and hence the associated downwelling/upwelling

4. Tidal mixing enhances the upwelling by mixing of water column (a local effect) and by advecting water mixed upstream (a remote effect)

5. River discharge inhibits the development of MKU through enhanced stratification

- The only general comment/doubt I have is that this study focuses on a single year of intense upwelling (summer of 2018). I wonder how representative the identified mechanisms are for other years. The narrative in the conclusions seem to imply this is the case. The role of the tides should be the same for other years, but the details regarding the convergence/divergence of currents could be more variable, both in intensity and perhaps in localization, despite the fact that topographic steering plays an important role.

[Figure]

*Figure A : Normalized yearly upwelling index (a) and JJAS averaged wind (b) between summers 2009 and 2018 in the LONG simulation. From To-Duy et al (2022, their Figure 13).*

To-Duy et al. (2022) showed that wind over the SVU region was weaker than average during summer 2017, and so was the upwelling intensity (see Fig. A above, extracted from To-Duy et al. 2022). 2017 and 2018 therefore represent two cases representative of respectively weak and strong summer wind and upwelling. To discuss the representativeness of our conclusions obtained from the analysis of summer 2018, we thus examined the variability of the upwelling development over summer 2017, also

simulated in the FULL, NoTide and NoRiver 2-year ensembles. Figure B below shows for summers 2017 and 2018 the maps of summer SST in the three ensembles. Figure C shows the daily time series of wind over the SVU region, and of the ensemble average $UI_{d,box}$ and intrinsic variability $VI_{d,box}$ of daily upwelling intensity over its four areas of development. The correlation between wind over the SVU region and $UI_{d,box}$ in the FULL ensemble is provided in Table A.

Analysis of summer 2017 confirms the conclusions obtained from the analysis of summer 2018 :
- The intraseasonal chronology of upwelling intensity for OFU, SCU and MKU is primarily driven by wind (Fig. Ca,b,d,h), with highly significant correlations (at more than 99%) between $UI_{d,box}$ and wind for both summers 2017 and 2018 (between 0.55 and 0.68, see Table A).
- The intraseasonal variability of upwelling intensity of NCU is not driven by wind (see Fig. Ca,f and the not significant correlations between $UI_{d,box}$ and wind for both summers 2017 and 2018, Table A). As for summer 2018, NCU only develops during summer 2017 at the beginning and end of summer, confirming our conclusion about the blocking role of the general circulation that prevails over the area during the core of summer.
- For both summers 2017 and 2018, the influence of OIV on upwelling intensity is very weak for MKU, weak for SCU, and stronger for OFU and NCU (Fig. Cc,e,g,i). The stronger influence of OIV on OFU and SCU in 2017 compared to 2018 is presumably related to smaller values of upwelling intensity (since VI is computed as the ratio between the ensemble standard deviation and average).
- As already observed for summer 2018, summer 2017 shows no significant impact of tides and rivers neither on ensemble average intensity nor on OIV of SCU, OFU and NCU (Fig. Be,f and Fig. Cb-g)
- Tides have a major role in MKU development both for 2017 and 2018, with no MKU developing at all in the NoTide ensemble for summer 2017 (Fig. Be and Ch), and rivers slightly reducing the upwelling intensity in the middle of summer.

Figure D below moreover shows the yearly upwelling location, materialized by the $UI_y$ 0.2°C iso-contours, and for summers 2017 and 2018 of the FULL ensemble and also for summers 2009-2018 of the LONG simulation evaluated and analysed by To-Duy et al. (2022). For summer 2018, the 10 members show rigorously the same location of MKU development. This area is strongly reduced for summer 2017. Over the 2009-2018 period, MKU always develops over the same core area, with a spatial extension varying with the strength of the upwelling. Following the comment of another reviewer, we also included in the paper a discussion about the role of topography, performing an additional simulation where the topography over the MKU shelf is smoothed (Section 4.1, page 12, lines 353-360, and Figure 7 of the revised manuscript). This change in topography results in a change of the southern limit of MKU extension. Those results confirm that the stability of the location of MKU development, indeed related to the influence of topography.

This further analysis of summer 2017 in our three ensembles and of the 2009-2018 simulation therefore suggests that our conclusions based on the detail analysis of summer 2018 regarding the mechanisms involved in the development and intraseasonal variability of SVU (wind, general circulation, intrinsic variability, tides, rivers and topography) over its four area of development, and in particular over the MKU region, are robust throughout the different years and associated atmospheric, oceanic and river conditions.

→ following this comment, we added those figures (Figures 2, 11,12) and a whole dedicated section (*Section 5, Representativeness of summer 2018,* page 16) in the revised version of our manuscript and modified the Introductioon (page 3, lines 110-111 ) and conclusion accordingly (page 17, lines 532-534).

*Table A : correlation between the daily wind averaged over the whole SVU area and the ensemble average of UIy is provided in Table A in the FULL ensemble. Correlation significant at more than 99% (at less than 90%) are in bold (italics).*

|  | SCU | OFU | NCU | MKU |
|---|---|---|---|---|
| Summer 2017 | **0.594** | **0.554** | *-0.167* | **0.621** |
| Summer 2018 | **0.596** | **0.604** | *-0.025* | **0.683** |

[Figure]

*Figure B : Ensemble average SST over June-September 2017 in the FULL (a), NoTide (b) and NoRiver (c) ensembles and difference between the NoTide (d) and NoRiver (f) and FULL ensembles (∘C). Panel d shows the bathymetry of the domain (m). Color bars for panels (a-c) and (e-f) are provided on the top and bottom right, and color bar for panel (d) on the bottom left.*

[Figure]

*Figure C : Daily time series over summers 2017 of averaged wind stress (a, N.m−2) over the whole upwelling region (7.5-14°N, 106-114°E) and of the ensemble mean of $UI_{d,box}$ and of $IV_d(UI_{d,B})$ (∘C) for the FULL (black), NoTide (green) and NoRiver (blue) ensembles for NCU (b,c), SCU (d,e), OFU (f,g) and MKU (h,i). Shaded green and blue colors shows the areas where the difference between the reference FULL and sensitivity NoTide and NoRiver ensembles is statistically significant at more than 99%.*

[Figure]

*Figure D : isolines of 0.2°C of yearly upwelling index UIy for the ten summers (June-September) 2009 to 2018 in the LONG simulation (left), and for summer 2017 (middle) and 2018 (right) in the ten members of the FULL ensemble. The difference between ensemble average of summer sea surface salinity (SSS) in the FULL and NoRiver ensembles is also showed in the right panel to highlight the Mekong river plume position.*

- I also think that providing some model validation could be useful. Is the upwelling well represented by the model when confronted by observations?. Is the MKU upwelling observed by the satellite SST products? If so, describing this could help the reader to assess how representative the mechanisms described here are.

The evaluation of the ability of the model to represent ocean dynamics and water masses from daily to interannual scales and from coastal to regional areas was demonstrated in details in the study of *To-Duy et al. (2022).* They compared a simulation performed over the period 2009-2018 with satellite datasets (sea surface temperature (OSTIA and JAXA), salinity (SMOS) and elevation (AVISO)) and with four in-situ datasets (temperature and salinity profiles of ARGO floats over 2009-2018, glider data over 2017 and CTD data during September 2018; SST and SSS from TSG data during summer 2014). In particular their high-resolution simulation, together and in agreement with a careful examination of available satellite and in situ data, showed for the first time the existence of MKU, and the ability of the model to represent it. Limitations associated with classically used gridded satellite data indeed strongly reduce the spatial observability of small coastal areas.

→We added a paragraph in the text to refer to the study of *To-Duy et al. (2022)* (Section 2.1, page 5, lines 128-134).

- One additional note: Fig. 9 nicely summarizes the findings described in the manuscript. Thank you!. However it seems to me that such Figure is not referenced in the manuscript (or maybe I missed it).

Indeed, thank you for pointing this out ! We refer to this figure in the revised manuscript (Conclusion, page 17, lines 533-534).

**Specific comments**

- P4,L17. It is mentioned that the simulation covers 2017-2018. However just the summer of 2018 seems to be analyzed. Why? Please clarify

We performed a first study about the South Vietnam Upwelling at the interannual scale over the period 2009-2018, published in *To-Duy et al. (2022).* This study showed that the upwelling was particularly strong during summer 2018, due to very strong southwest monsoon wind during this summer. Summer 2018 was therefore chosen as a case study by *Herrmann et al. (2023)*, and in the following manuscript. However, as pointed out by the reviewer, our ensemble simulations also cover the period 2017, which was a summer of weak upwelling for the four boxes in the 2009-2018 simulation of *To-Duy et al. (2022)*. We therefore examined summer 2017 to examine the robustness of our conclusions regarding the effects of tides and rivers on the upwelling strength and OIV over its four areas of development, and

regarding the functioning of MKU. This is explained in details in the answer to the first general comment above.

- P4: since Kz is reported later in the analysis of the sensitivity to river discharge experiment, it could be good to mention what vertical mixing parameterization is used in the model

Vertical mixing is parameterized according to the k-epsilon turbulence closure scheme (*Rodi, 1987*). This was added in the revised manuscript (Section 2.1, page 4, lines 116-117).

- P5: why MKU does not include the coast?

The four upwelling boxes were defined by *To Duy et al. (2022)* based on the spatial distribution of SST and upwelling spatial index $UI_{JJAS}$, and with the aim to fully cover the upwelling development areas. We show in Fig. D above the 0.2°C isoline of $UI_{JJAS}$ for the 10 summers (JJAS) of the 2009-2018 simulation examined by *To Duy et al. (2022),* and for the 10 members of summers 20187 and 2018 of the FULL ensemble. They confirm that MKU does not develop along the coast, but slightly off the Mekong delta coastline, and that the MKU box designed by *To Duy et al. (2022)* cover the upwelling development offshore the Mekong delta coastline. Note that we can see small coastal area of low SST at the Mekong mouth, but we excluded those small areas since our model ~1km resolution along the coast as well as the coarse prescription of river water temperature (28°C) may not be perfectly suited to represent the estuarine dynamics. One could include the coast but this would not change a lot the results given the very small size of those areas.

→ Following this comment, we added Figure 2 in the revised manuscript, and commented it in the revised manuscript (Section 2.3.1, page 6, lines 174-178).

- P7: in "SEJ and eddy dipole are well established in July-August, preventing NCU to develop, and much weaker at the beginning (June) and end (September) of summer, allowing NCU to develop." A similar effect is observed for the OIV, could you comment on this?

*Herrmann et al. (2023)* showed that NCU is first driven by the large scale summer circulation, that prevents or allows its development. When the cyclonic circulation off the NCU region related to the summer dipole and SEJ that prevails is well established, NCU can not develop. As a result, its chaotic variability is weak: it is weak whatever the member of the ensemble. Conversely, when the cyclonic circulation is weak, at the beginning and end of summer, NCU development is allowed depending on the organization of small scale circulation over the NCU area, which is strongly chaotic. NCU OIV is therefore strong during the periods of allowed development, and weak during the periods of prevented development, explaining the similar chronology of NCU intensity and OIV.

→ We added a few lines in the revised manuscript to comment this (Section 3.1, page 8, lines 224-232).

- In "Strong mid-July and mid-August wind peaks indeed induce peaks of upwelling intensity for MKU, SCU and OFU in the FULL ensemble analyzed in this previous study, while NCU does not develop during the core of the summer but in late June and late August (Fig. 2)." Could you comment on how different is the wind stress for other years?. That is, I wonder how representative these results are for other years.

*To-Duy et al. (2022)* already showed from the analysis of their 2009-2018 simulation that the development of NCU is inhibited or favored depending on the circulation that prevails over BoxNC. They showed that the NCU is inhibited when alongshore currents, either southward or northward and resulting from two opposite situations in terms of wind and offshore circulation, prevail over BoxNC. Southward currents are associated with the general cyclonic circulation that prevails offshore BoxNC during summers of strong wind over the SVU region, as for summer 2018, while northward alongshore currents can only develop during years of weak wind. *To-Duy et al. (2022)* showed that NCU can only develop during periods of weak general circulation, at the beginning and end of summer or during summers of very weak wind, allowing the circulation prevailing over BoxNC, which is highly chaotic, to be offshore oriented. The analysis of summer 2017 detailed above (Fig. Cf) further confirms those conclusions, showing a development of NCU at the beginning and end of summer and no NCU during the core of summer.

→ We added a few lines in the revised manuscript to comment this (Section 3.1, page 8, lines 224-232 and also Section 5, page 16, line 486-488).

- P7: In "The effect of river discharge on the intrinsic variability of in MK is much weaker :" This is a bit counter-intuitive. River discharge will introduce strong buoyancy gradients in the mixed layer, which promote the development of submesoscale variability. Is this because BoxMK excludes the coastal region?

We indeed explained above that the area of upwelling development over the shelf does not reach the very coastal region, which is therefore not included in BoxMK. We show in Fig. D above the difference of sea surface salinity (SSS) averaged over JJAS in the FULL and NoRiver simulation: the area where this difference is significant (exceeding ~1 psu) highlights the area of influence of the Mekong River plume. Fig. D indeed shows that the plume hardly reaches the MKU box, only covering its northwestern corner, as already shown for July 31$^{st}$ in Fig. 10a of the manuscript. Finally river discharge therefore only affects the vertical stratification over a small area of boxMK (near point G, Fig. 10a,c). The area where strong buoyancy gradients could develop and enhance chaotic variability is thus not included in BoxMK, or very small, explaining that river discharge do not have a significant influence on MKU OIV.

→ We added a few lines in the revised manuscript to comment this (Section 4.3, page 15, lines 471-476 and Fig. 10ac).

- P8: In " We take M17, i.e. the simulation where both the small and large scale states are given by conditions of January 2017 of the PSY4QV3R1 analysis", Is there any validation of the model solution over this period? At least for the SST, SLA and maybe SSS ... If it was reported previously, it would be good to reference that.

See answer above for the general comment : we added a paragraph in the text to refer to the study of *To-Duy et al. (2022)* who carefully compared the model with satellite SST, SLA and SSS data as well as in-situ T and S profiles, from the intraseasonal to the interannual scales (Section 2.1, page 5, lines 128-134).

- P9: In "the edge of the continental slope, this barotropic warm northeastward current meets a cold bottom northwestward current that flows from the open area towards the coast (Figs. 3d,4d)" Is this really at the edge of the continental slope?

Indeed, this is more generally on the shelf slope. The position of this convergence is driven by the topography, as suggested by an analysis done to answer a comment of the other reviewer about the influence of topography.

→ We changed "the edge of the continental slope" by "on the shelf slope" in the text (Section 4.1, page, line) and added a whole paragraph about the influence of topography (Section 4.1, page 11, line 335).

- P13: In "This stratification weakening makes the water column easier to mix vertically, facilitating the tidal vertical mixing, which is the main contributor to MKU in this area and during the transition period, as explained above" Is there any extra contribution due to wind forced coastal upwelling in the NoRiver case?

One could indeed expect that the weaker stratification along the coast could indeed favor the coastal Ekman transport. Examining carefully the SST difference between FULL and NoRiver over the MKU region, there is however no significant intensification of MKU that could be induced by wind forced upwelling intensification (Fig. Bf), neither for 2017 nor for 2018. The river effect on the stratification is therefore not sufficient at the coast to significantly enhance the Ekman transport driven upwelling.

→ We added a few lines in the revised manuscript to comment this (Section 4.3, page 15, line 471-476)

- P13: In "Three sensitivity", the FULL experiment is the reference state. Therefore two sensitivity experiments were performed (NoTide and NoRiver).

Indeed ! This has been modified accordingly in the Introduction (page 4, lines 104 and 118) and conclusion (page 16, line 504).

**Technical corrections**

Note: English is not my first language. Therefore non-typo corrections are just suggestions to be taken with a (huge) grain of salt.

We are not English native speakers ourselves. In order to improve our writing and to answer the comments below, we therefore also asked for the advice of an English native speaker colleague.

- P1: Maybe change "investigating" by "investigates"?

This has been modified accordingly (page 1, line 5).

- P2: In "~ 14°E",  is this 14 deg N?

This has been corrected (page 2, line 30).

- P2, L56: change "and" by "are"?

This has been corrected (page 2, line 57).

- P3,L67: Change "tidal" by "the tidal"?

This has been corrected (page 3, line 68).

- P4: Maybe change "ensemblist" by "probabilistic"?

This has been modified accordingly (page 4, line 101).

- P4:L108: Change "refsec:conclusion" by 5.

This has been corrected (page 4, line 111).

- P7: "n :"  There are numerous spaces before punctuation colons. I wonder if this was just an effect of the pdf rendering …

This has been corrected throughout the manuscript.

- P11: Change "°C" by "deg E"

This has been corrected (page 14, line 416).

- P12: In "tidal mixing is a major factor of MKU maintaining." Maybe "on the maintenance of MKU"?

We replaced by "the contribution of the horizontal circulation gradient to MKU is therefore strongly weakened, and tidal mixing is a major factor in maintaining MKU" (page 13, line 434).

- P12, L360: "in" by "on"?

Apparently "in" seems correct.

- P12: In "fresh hence light .." maybe "fresher (and hence lighter) water .."`

This has been modified accordingly (page 14, line 456).

**References**

Herrmann, M., To Duy, T., and Estournel, C.: Intraseasonal variability of the South Vietnam upwelling, South China Sea: influence of atmospheric forcing and ocean intrinsic variability, Ocean Science, 19, 453–467, https://doi.org/10.5194/os-19-453-2023, 2023.

Rodi, W., 1987. Examples of calculation methods for flow and mixing in stratified fluid. J. Geophys. Res. 92, 5305–5328.

To Duy, T., Herrmann, M., Estournel, C., Marsaleix, P., Duhaut, T., Bui Hong, L., and Trinh Bich, N.: The role of wind, mesoscale dynamics, and coastal circulation in the interannual variability of the South Vietnam Upwelling, South China Sea – answers from a high-resolution ocean model, Ocean Science, 18, 1131–1161, https://doi.org/10.5194/os-18-1131-2022, 2022.

---

## Author Comment (AC2)

Reviewer 2

We warmly thank the reviewer for the time and attention devoted to our paper, and for those positive and constructive comments. We have carefully considered all his comments and suggestions in the revised version of our manuscript. In what follows, our answers and modifications are highlighted in blue. Line numbers refer to the highlighted version of the revised manuscript.

This article describes the authors' analysis of an ensemble of simulations to analyze the SST variability in the Gulf of Thailand off the coast of Vietnam. They use a ten-member ensemble computed with a high-resolution ocean model to compare the impacts of eddies, tides, and river runoff during a two-year period. They find that the SST, which they characterize in terms of an "upwelling index", is mostly determined by winds, but in the Mekong Delta region they find that tides alter the currents and vertical turbulent heat transport enough to significantly influence the SST. This paper should be of interest to oceanographers and fisheries scientists interested in the detailed analysis of this region.

Overall, I found the paper well-written and logically laid out. My main questions and suggestions for improvement are as follows:

- I found the initial characterization of "upwelling" in terms of SST confusing. I think readers will be confused because upwelling normally refers to vertical velocity, but it is clear that it is really SST which is the focus of this article. I understand that this may be done because they want to use language and diagnostics consistent with previous analyses of this area, but it would be helpful to be explicit about this distinction.

This paper indeed belongs to an ensemble of studies, from the same authors and from other colleagues, that investigated the functioning and variability of upwelling that develops off the Vietnamese coast, in particular through its signature of SST (*Xie et al. 2003, 2007, Da et al. 2019, Ngo and Hsin 2021, To-Duy et al. 2022,* Herrmann et al. 2023 and many others cited in the paper). Our paper specifically aims at understanding the intraseasonal variability of the South Vietnam upwelling over its four areas of development, investigating in particular the effects of tides and rivers, and exploring the mechanisms involved in MKU functioning, that was revealed by *To-Duy et al. (2022)*. We indeed wanted to be consistent with the language and diagnostics of those previous studies, hence using SST as an indicator of upwelling intensity. As explained by *Da et al. (2014)*, upwelling indicators can be built based on surface wind (Ekman transport theory) or SST (see *Benazzouz et al., 2014*, for a review). The advantages of an SST-based indicators are, first, that they provide information on the upwelling intensity but also on the spatial distribution of upwelled water that reaches the surface and triggers primary production and, second, that it can be applied on real SST data derived from satellite observations to monitor the upwelling from observational data. To better understand the mechanisms involved in the upwelling functioning over the Mekong shelf, that was not studied until now, we also characterize the upwelling in terms of vertical velocity in part *4. Circulation mechanisms* of the paper).

→ Following this comment, we explained more clearly that we base our study of the upwelling on SST indicators (Introduction, page 4, lines 95-96 and Section 2.3, page 5, lines 150-154 )

- (2) Because there is no discussion of degrees of freedom, I don't think that the t-test and F-test estimates of statistical significance are justified or useful. The discussion also mentions correlations between the FULL NoTides and NoRivers simulations, and I think this is a perfectly acceptable qualitative characterization of the results instead.

To quantify the effect of tides and rivers on the upwelling intensity and intrinsic variability, we indeed compute the relative differences $\Delta_m(UI_d)$ and $\Delta_\sigma(UI_d)$ of the mean (that quantifies the intensity) and

standard deviation (that quantify its intrinsic variability) of the 10-member vector of yearly upwelling index $UI_{JJAS}$ between the FULL reference simulation and in the NoRiver or NoTide sensitivity simulation, as explained in section 2.3.3. However, their sole values, reported in Table 1, do not allow to objectively estimate if those differences, hence the effect of tides or rivers, are significant or not. This is why, following Da et al. (2019), we compute the p-values $p_m$ and $p_\sigma$ associated respectively with the t test (for the mean difference) and F test (for the standard deviation difference): $p_m$ and $p_\sigma$ provide the degree of significance of those differences. We apply those tests on the 10-member vectors of $UI_{JJAS}$ and in the reference and sensitivity simulation and report them in Table 1. We also compute $p_m$ and $p_\sigma$ to assess the significance of difference of daily upwelling index $UI_d$ in Figure 2 of the manuscript, highlighting in colors the periods when the differences are significant at more than 99% (p-value<0.01).

In the paper, we also investigate the relationship between the chronology over JJAS of several variables by computing the Pearson correlation coefficient and associated p-values (that again quantify the statistical significance of the correlation) between their 122-day time series.

→ We explained that more clearly in the revised manuscript, writing in particular that t and F tests are performed over 10-member size vectors, and that correlations are computed between 122-day size time series, which therefore inform about the degree of freedom (Section 2.3.3, page 7, lines 192-211). We moreover simplified the explanations of OIV indicator in part 2.3.2, replacing the generic "X" variable by the $UI_d$ and $UI_{JJAS}$ variables over which we actually perform the computation at the daily and summer (JJAS) scales.

- (3) Because the results find, essentially, that the tides influence Mekong upwelling, and the other regions are not much affected, I think the other regions could mostly be left out of the discussion. Perhaps the discussion of the other regions could be emphasized only in the introductory material. Similarly, in the abstract, consider re-ordering the presentation so that the positive or significant results are stated first, and the no-impact results are stated as a contrasting follow-up.

Since our study focuses in the SVU, since previous studies questioned the influence of tides and rivers on SVU areas other than MKU, and given the questions of the other reviewers, we chose to keep the results concerning the effect of tides and rivers on OFU, NCU and SCU. We however followed the suggestion of the reviewer by reorganizing the Abstract, Part 3 (*Impact of tides and rivers in the four upwelling areas*) and Conclusion, reducing as much as possible the non significant results, and presenting the significant results first (see Abstract, page 1, lines 11-19 ; Section 3, pages 8-10, lines 221-300 and Conclusion, pages 16-17, lines 506-516).

- (4) I think the article would be more impactful if it emphasized the analysis of the term balance of the temperature evolution equation. You could map the Simpson-Hunter number (the ratio of surface heat flux to tidally-generated vertical turbulent heat flux) to delineate the regions where you would might expect tidal currents to be significant in the SST budget. Since you mention the significance of locally-created vs non-local (horizontally-advected) stratification, could you quantify this by mapping an appropriate non-dimensional number. Likewise, in the analysis of divergence you highlight the role of topographic features. This could be captured by comparing bottom u*grad(H) vs. H*div(u) at mid-depth.

- To highlight the significance of locally-created vs non-local (i.e. horizontally-advected) surface cooling, we examine the maximum over the water column of the difference of $\log_{10}(Kz)$ between FULL and NoTide, $\Delta log_{10}(Kz)$, where Kz is the vertical diffusivity coefficient. $\Delta log_{10}(Kz)$ quantifies the vertical mixing induced by tides: a value higher than 2 means that Kz is $10^2$ higher with tides than without tides, i.e. that the vertical mixing induced locally by tides dominates the vertical mixing (see profiles of temperature and Kz in Figure 9 of the paper). Conversely, a value lower than 1 means that tides do not significantly contribute to the local vertical mixing. We show $\Delta log_{10}(Kz)$ on figure Ad below, together

with the maps of SST in FULL and NoTide and of their difference. We also indicate the -0.4°C contours of the SST difference between FULL and NoTide : it highlights the area of surface cooling induced by tides. High values of $\Delta log_{10}(Kz)$ in this area *(>2)* corresponds to tidally induced surface cooling due to local tidal vertical mixing (e.g. points F, E, D, as shown in Fig. 9 of the paper). Low values (<1) correspond to tidally induced surface cooling due to horizontal advection of water mixed upstream (e.g. point A downstream point F).

→ we added this in the revised version of the paper (Section 4.2.2, pages 14-15, lines 440-4476 and, Figure 9)

[Figure]

*Figure A : Maps of SST on 16/07/2018 in M17 of the FULL ENSEMBLE (a), of the NoTide ensemble (b) and their difference (c), and map of $\Delta log_{10}(Kz)$ on the same day plotted only for areas where the SST difference exceeds 0.4°C. The black line shows the isotherm of To=27.6°C that corresponds to the region of upwelling occurrence.*

- To highlight the effect of topography, we plot below in Figure B the vertical velocity induced by the topography gradient, $u_{bottom}$ . grad(h) (Fig. Ba), and the ratio between $u_{bottom}$ . grad(h) and the surface velocity (Fig. Bb), following the suggestion of the reviewer. We can see spots of strong values of topography induced vertical velocity over places of steep topography, in particular some small sea mounts (see Fig. Ca that shows the bathymetry), suggesting and effect of topography. This effect is however not really highlighted by the ratio figure. In order to highlight the role of topography in a more convincing way, we therefore performed an additional simulation (with rivers and tides) where we smoothed those topographic anomalies, and compare the resulting vertical velocity with the FULL ensemble but also the LONG simulation performed with the same model configuration over the period 2009-2018 by *To-Duy et al. (2022).*

[Figure]

*Figure B : maps of bottom vertical velocity induced by topography gradient ($u_{bottom}.grad(h)$, m.s$^{-1}$, left) and ratio between $u_{bottom}.grad(h)$ and the surface vertical velocity (right).*

We show in Fig. Ce-h below the surface vertical velocity on July 16$^{th}$, 2018 for 2 members of the FULL ensembles (the other members, not show, are extremely similar), for the LONG simulation and for the simulation with smoothed bathymetry. The 10 members of the FULL ensemble, but also the LONG simulation of *To-Duy et al. (2022)* show extremely similar positions and values of strong surface velocity (see dashed line on Fig. Cd-e). In the simulation with the smoothed bathymetry, the positions of strong vertical surface velocity change, with the line of strong positive vertical velocity near the dashed line shifted by ~30 km to the west. Modifying the topography, removing in particular the small seamounts, therefore modifies in this sensitivity test the upwelling location that hardly changes varies within the FULL ensemble and in the LONG simulation, confirming the determining role of topography in this location.

→ We added this in the revised version of the paper (Section 4.1, page 12, lines 352-360, and Figure 7).

[Figure]

*Figure C: Initial bathymetry (a, m) and smoothed bathymetry (b), the dashed ellipse shows the area of bathymetry smoothing. Surface vertical velocity (m.s⁻¹) on 16/07/2018 in M09 (c) and M18 (d) of the FULL ensemble (the other members, not shown, are extremely similar), in LONG (e) and in the sensitivity simulation with smoothed bathymetry (f). The dashed line highlights the front of strong upward vertical velocity.*

- Investigating into details the heat budget over the area, quantifying the role of the relative contributions of atmospheric fluxes, lateral transport and vertical advection and mixing is indeed a question that we plan to address. We show in Fig. A the daily timeseries atmosphere net heat flux over the region: a few periods of negative heat fluxes, i.e. inducing ocean surface cooling, can be observed during some wind peak events. Those periods are however very short with a heat loss hardly exceeding 50 W.m$^{-2}$. Moreover, though the NoTide simulation is submitted to a very similar atmospheric heat flux, it produces a much weaker upwelling as explained in details in the paper. This suggests that the effect of atmospheric cooling is much weaker than the effect of vertical velocity and of vertical tidal mixing. We could answer in details to this question using the online closed water, heat and seat budget tool available in the mode, as we did in the study of Trinh et al. (2024) with a 4 km resolution configuration of SYMPHONIE over the South China Sea to examine the relative contributions of lateral fluxes at interocean straits, river and atmospheric forcing and internal variations in the seasonal variability of those budgets. Beside, the effect of the upwelling and associated surface cooling to the atmosphere has also been mentioned in the literature, in particular its effect on sea breeze winds (*Zheng et al. 2016, Yu et al. 2020*). We plan to investigate this question with the ocean-atmosphere coupled SYMPHONIE-RegCM model recently developed in the framework of the *Quentin Desmet (2024)* PhD to study air-sea coupled interactions in the region. As explained above, the present paper really focuses on the upwelling functioning and variability, that we quantify and explain based on SST, but also on horizontal and vertical velocity. The study of heat (as well as water and salt) budget would not only require additional simulations, we moreover think that we would deserve a dedicated analysis and paper.

→We therefore choose not to investigate the question of air-sea heat fluxes here, but mention this question in the manuscript (Conclusion, page 17-18, lines 540-547).

- (5) There is no discussion to justify the randomization technique used to create the ensemble. The approach taken seems perfectly reasonable, but it would be good to discuss why the 100km cutoff between large-scale and small-scale is appropriate, and also to mention why other sources of randomness (such as the winds or large-scale stratification) were not used.

Indeed, in the submitted version of the manuscript, we only referred to *Herrmann et al. (2023)* where the ensemble creation strategy was explained. This strategy is based on the fact that most of the OIV develops at (sub)mesoscale, related in particular to the presence of eddies of strongly chaotic behavior. *Herrmann et al. (2023)* indeed explained: "*For that we used ten different initial conditions for temperature, salinity, sea surface elevation and currents fields. Most of the OIV develops at mesoscale (Sérazin et al., 2015, Waldman et al., 2018), we therefore only perturbed the mesoscale field, following the same methodology as Waldman et al. (2017a, 2017b, 2018). For the ten simulations of the ensemble, the large-scale state of the initial field is identical, and the small-scale of the initial field state differs. The common large-scale state is equal to the large-scale state of January 1st, 2017 of the LONG simulation, computed using a 100 km low-pass filter. For XX going from 09 to 18, the small-scale state of January 1st, 20XX of the LONG simulation is computed using a 100 km high-pass filter.*" The 100 km cutoff was therefore chosen to separate the large scale circulation from the (sub)mesoscale processes, which develops at scale smaller than 100 km (*Lin et al. 2020, Ni et al. 2021*).

Indeed, as pointed out by the reviewer, applying perturbation on the atmospheric fields that drive the upwelling (i.e. the wind, see for example *Nguyen-Duy et al. 2023*) or to the lateral boundary conditions that would create a different circulation fields at the submesoscale to mesoscale (see for example *Da et al. 2019*) would also introduce chaoticity in our simulations, and the study of their effect would be of interest. Perturbing the large scale stratification could also be a source of chaoticity, but would also represent factors other than the sole chaoticity effect, in particular the effect of interannual to long term atmospheric conditions and lateral oceanic forcing, making the sensitivity tests difficult to interpret as a result of chaoticity alone.

→ We added a more detailed explanation about the perturbation strategy in the revised version of the manuscript, mentioning also the role of other perturbing factors as winds and lateral boundary conditions (Section 2.2, page 5, lines 139-142 and Conclusion, page 18, lines 548-550).

Overall, I think this careful analysis will be of interest to researchers studying this region. I recommend publication after minor revision, if the authors choose; or they might wish to pursue the more extensive revision implied by item (4), above.

As explained above, we chose to examine the questions rose by the reviewer in (4), in particular regarding the relative local vs. remote effects of tides on sea surface cooling, and the role of topography.

[Dear authors and editors:

I spent the last 2.5 hours reading this manuscript and enumerating my detailed comments in this text box. Unfortunately, when I clicked "Intermediate save" the website asked me to authenticate again, and all of my comments were lost when this web page again re-opened. I am not willing to re-create my detailed comments. I have re-created my general comments, above.]

**Referencces**

Benazzouz, A., Mordane, S., Orbi, A., Chagdali, M., Hilmi, K., Atillah, A., et al. (2014). An improved coastal upwelling index from sea surface temperature using satellite-based approach—The case of the Canary Current upwelling system. Continental Shelf Research, 81, 38–54. https://doi.org/10.1016/j.csr.2014.03.012

Da, N. D., Herrmann, M., Morrow, R., Ni.o, F., Huan, N. M., and Trinh, N. Q.: Contributions of Wind, Ocean Intrinsic Variability, and ENSO to the Interannual Variability of the South Vietnam Upwelling: A Modeling Study, Journal of Geophysical Research: Oceans, 124,6545–6574, https://doi.org/10.1029/2018JC014647, 2019.

Desmet, Q. : Exploring the keys to advance air–sea coupled regional modeling for deeper insights into Southeast Asian climate. Université de Toulouse, PhD thesis, 2024.

Herrmann, M., To Duy, T., and Estournel, C.: Intraseasonal variability of the South Vietnam upwelling, South China Sea: influence of atmospheric forcing and ocean intrinsic variability, Ocean Science, 19, 453–467, https://doi.org/10.5194/os-19-453-2023, 2023.

Lin, H.; Liu, Z.; Hu, J.; Menemenlis, D.; Huang, Y. Characterizing Meso- to Submesoscale Features in the South China Sea. Progress in Oceanography, 188, 102420. https://doi.org/10.1016/j.pocean.2020.102420, 2020

Ngo, M. H. and Hsin, Y. C.: Impacts ofWind and Current on the Interannual Variation of the Summertime Upwelling Off Southern Vietnam in the South China Sea, J. Geophys. Res.-Ocean., 126, e2020JC016892, https://doi.org/10.1029/2020JC016892, 2021.

Nguyen-Duy, T., N. K. Ayoub, P. De-Mey-Frémaux, T. Ngo-Duc, How sensitive is a simulated river plume to uncertainties in wind forcing? A case study for the Red River plume (Vietnam), Ocean Modelling, Volume 186, 102256, ISSN 1463-5003, https://doi.org/10.1016/j.ocemod.2023.102256, 2023

Ni, Q.; Zhai, X.; Wilson, C.; Chen, C.; Chen, D. Submesoscale Eddies in the South China Sea. Geophysical Research Letters, 48 (6). https://doi.org/10.1029/2020GL091555, 2021

To Duy, T., Herrmann, M., Estournel, C., Marsaleix, P., Duhaut, T., Bui Hong, L., and Trinh Bich, N.: The role of wind, mesoscale dynamics, and coastal circulation in the interannual variability of the South Vietnam Upwelling, South China Sea – answers from a high-resolution ocean model, Ocean Science, 18, 1131–1161, https://doi.org/10.5194/os-18-1131-2022, 2022.

Xie, S.-P., Q. Xie, D. Wang, and W. T. Liu, Summer upwelling in the South China Sea and its role in regional climate variations, J. Geophys. Res., 108(C8), 3261, doi:10.1029/2003JC001867, 2003.

Xie, S.-P., C.-H. Chang, Q. Xie, and D. Wang , Intraseasonal variability in the summer South China Sea: Wind jet, cold filament, and recirculations, J. Geophys. Res., 112, C10008, doi:10.1029/2007JC004238, 2007

Yu, Y.,Wang, Y., Cao, L., Tang, R., and Chai, F.: The ocean-atmosphere interaction over a summer upwelling system in the South China Sea, Journal of Marine Systems, 208, 103-360, https://doi.org/10.1016/j.jmarsys.2020.103360, 2020.

Zheng, Z.-W., Zheng, Q., Kuo, Y.-C., Gopalakrishnan, G., Lee, C.-Y., Ho, C.-R., Kuo, N.-J., and Huang, S.-J.: Impacts of coastal upwelling off east Vietnam on the regional winds system: An air-sea-land interaction, Dynamics of Atmospheres and Oceans, 76, 105–115, https://doi.org/10.1016/j.dynatmoce.2016.10.002, 2016.

---

## Author Comment (AC3)

Reviewer 3

We warmly thank the reviewer for the time and attention devoted to our paper, and for those positive and constructive comments. We have carefully considered all those comments and suggestions in the revised version of our manuscript. In what follows, our answers and modifications are highlighted in blue. Page and line numbers refer to the highlighted version of the revised manuscript.

This manuscript presents an investigation of the effect of tides and rivers on upwelling intensity over upwelling areas in Vietnamese waters in the summer of 2018 at different time scales. In addition, the physical mechanism of Mekong upwelling development was further investigated. The results of this paper is very interesting, comprehensive, and deepen the understanding of upwelling and the dynamics of upwelling areas. The manuscript is quite fully done and well-constructed. I suggest the manuscript to be accepted after minor revisions. Here are my comments:

Some technical mistakes should be corrected:

- ":" should not have a space before it.

This has been corrected throughout the manuscript.

- Line 29: "14 E" should be "14 N"

This has been corrected (page 2, line 30).

- Line 56: "and" should be "are"

This has been corrected (page 2, line 57).

- Line 81-82: two "moreover", line 91-92: two "however": should use one word!

One "moreover" has been replaced with "also" (page 3, line 82), and one "however" has been with "thus" (page 4, line 94)

- Line 108: "refsec:conclusion"?

Indeed it should be "Section 5". This has been corrected (page 4, line 111).

- Line 173: "Fig.1e,f" should be "Figs. 1e,f"; similar to others!

This has been corrected, here and throughout the manuscript

- In section 2.1, the limits of four numerical domains should be added with longitudes and latitudes in the caption of Fig. 1 or in the main text. It is not so good to read if we always need to check the paper of To Duy et al. (2022) or Herrmann et al. (2023).

Coordinates of the upwelling areas are now provided in the caption of Fig. 1 of the revised manuscript.

- In Fig. 1: the lower limit of the color bar is 26oC so the center of the upwelling area shows a white (blank) color, this color bar should be extended!

Fig. 1 has been redone to avoid blanks (temperatures below 26°C appear as dark blue, as it should.

- The authors already simulated, showed, and emphasized the spatial differences of SST of only a representative year of 2018 (Fig. 1). Why did the authors discuss that representative year? Do you get similar conclusions for 2017?

[Figure]

*Figure A : Normalized yearly upwelling index (a) and JJAS averaged wind (b) between summers 2009 and 2018 in the LONG simulation. From To-Duy et al (2022, their Figure 13).*

To-Duy et al. (2022) showed that wind over the SVU region was weaker than average during summer 2017, and so was the upwelling intensity (see Fig. A above, extracted from To-Duy et al. 2022). 2017 and 2018 therefore represent two cases representative of respectively weak and strong summer wind and upwelling. To discuss the representativeness of our conclusions obtained from the analysis of summer 2018, we thus examined the variability of the upwelling development over summer 2017, also simulated in the FULL, NoTide and NoRiver 2-year ensembles. Figure B below shows for summers 2017 and 2018 the maps of summer SST in the three ensembles. Figure C shows the daily time series of wind over the SVU region, and of the ensemble average $UI_{d,box}$ and intrinsic variability $VI_{d,box}$ of daily upwelling intensity over its four areas of development. The correlation between wind over the SVU region and $UI_{d,box}$ in the FULL ensemble is provided in Table A.

*Table A : correlation between the daily wind averaged over the whole SVU area and the ensemble average of UIy is provided in Table A in the FULL ensemble. Correlation significant at more than 99% (at less than 90%) are in bold (italics).*

|             | SCU   | OFU   | NCU     | MKU   |
|-------------|-------|-------|---------|-------|
| Summer 2017 | 0.594 | 0.554 | *-0.167* | 0.621 |
| Summer 2018 | 0.596 | 0.604 | *-0.025* | 0.683 |

Analysis of summer 2017 confirms the conclusions obtained from the analysis of summer 2018 :
- The intraseasonal chronology of upwelling intensity for OFU, SCU and MKU is primarily driven by wind (Fig. Ca,b,d,h), with highly significant correlations (at more than 99%) between $UI_{d,box}$ and wind for both summers 2017 and 2018 (between 0.55 and 0.68, see Table A).
- The intraseasonal variability of upwelling intensity of NCU is not driven by wind (see Fig. Ca,f and the not significant correlations between $UI_{d,box}$ and wind for both summers 2017 and 2018, Table A). As for summer 2018, NCU only develops during summer 2017 at the beginning and end of summer, confirming our conclusion about the blocking role of the general circulation that prevails over the area during the core of summer.
- For both summers 2017 and 2018, the influence of OIV on upwelling intensity is very weak for MKU, weak for SCU, and stronger for OFU and NCU (Fig. Cc,e,g,i). The stronger influence of OIV on OFU and SCU in 2017 compared to 2018 is presumably related to smaller values of upwelling intensity (since VI is computed as the ratio between the ensemble standard deviation and average).
- As already observed for summer 2018, summer 2017 shows no significant impact of tides and rivers neither on ensemble average intensity nor on OIV of SCU, OFU and NCU (Fig. Be,f and Fig. Cb-g)

- Tides have a major role in MKU development both for 2017 and 2018, with no MKU developing at all in the NoTide ensemble for summer 2017 (Fig. Be and Ch), and rivers slightly reducing the upwelling intensity in the middle of summer.

Figure D below moreover shows the yearly upwelling location, materialized by the UI$_y$ 0.2°C iso-contours, and for summers 2017 and 2018 of the FULL ensemble and also for summers 2009-2018 of the LONG simulation evaluated and analysed by To-Duy et al. (2022). For summer 2018, the 10 members show rigorously the same location of MKU development. This area is strongly reduced for summer 2017. Over the 2009-2018 period, MKU always develops over the same core area, with a spatial extension varying with the strength of the upwelling. Following the comment of another reviewer, we also included in the paper a discussion about the role of topography, performing an additional simulation where the topography over the MKU shelf is smoothed (Section 4.1, page 12, lines 353-360, and Figure 7 of the revised manuscript). This change in topography results in a change of the southern limit of MKU extension. Those results confirm that the stability of the location of MKU development, indeed related to the influence of topography.

This further analysis of summer 2017 in our three ensembles and of the 2009-2018 simulation therefore suggests that our conclusions based on the detail analysis of summer 2018 regarding the mechanisms involved in the development and intraseasonal variability of SVU (wind, general circulation, intrinsic variability, tides, rivers and topography) over its four area of development, and in particular over the MKU region, are robust throughout the different years and associated atmospheric, oceanic and river conditions.

→ following this comment, we added those figures (Figures 2, 11,12) and a whole dedicated section (*Section 5, Representativeness of summer 2018,* page 16) in the revised version of our manuscript and modified the Introductioon (page 3, lines 110-111 ) and conclusion accordingly (page 17, lines 532-534).

[Figure]

*Figure B : Ensemble average SST over June-September 2017 in the FULL (a), NoTide (b) and NoRiver (c) ensembles and difference between the NoTide (d) and NoRiver (f) and FULL ensembles (∘C). Panel d shows the bathymetry of the domain (m). Color bars for panels (a-c) and (e-f) are provided on the top and bottom right, and color bar for panel (d) on the bottom left.*

[Figure]

*Figure C : Daily time series over summers 2017 of averaged wind stress (a, N.m−2) over the whole upwelling region (7.5-14°N, 106-114°E) and of the ensemble mean of $UI_{d,box}$ and of $IV_d(UI_{d,B})$ (∘C) for the FULL (black), NoTide (green) and NoRiver (blue) ensembles for NCU (b,c), SCU (d,e), OFU (f,g) and MKU (h,i). Shaded green and blue colors shows the areas where the difference between the reference FULL and sensitivity NoTide and NoRiver ensembles is statistically significant at more than 99%.*

[Figure]

*Figure D : isolines of 0.2°C of yearly upwelling index UIy for the ten summers (June-September) 2009 to 2018 in the LONG simulation (left), and for summer 2017 (middle) and 2018 (right) in the ten members of the FULL ensemble. The difference between ensemble average of summer sea surface salinity (SSS) in the FULL and NoRiver ensembles is also showed in the right panel to highlight the Mekong river plume position.*

- Line 261: "It belongs to the large scale cyclonic circulation…" => I could not see this cyclonic circulation, is it not shown in Fig. 3 or it is anticyclonic?

This was indeed a typo, the correct work is anticyclonic as suggested by the reviewer, and has been corrected (p 11, line 325)

**References**

Herrmann, M., To Duy, T., and Estournel, C.: Intraseasonal variability of the South Vietnam upwelling, South China Sea: influence of atmospheric forcing and ocean intrinsic variability, Ocean Science, 19, 453–467, https://doi.org/10.5194/os-19-453-2023, 2023.

To Duy, T., Herrmann, M., Estournel, C., Marsaleix, P., Duhaut, T., Bui Hong, L., and Trinh Bich, N.: The role of wind, mesoscale dynamics, and coastal circulation in the interannual variability of the South Vietnam Upwelling, South China Sea – answers from a high-resolution ocean model, Ocean Science, 18, 1131–1161, https://doi.org/10.5194/os-18-1131-2022, 2022.